# Hypothalamic melanin-concentrating hormone neurons integrate food-motivated appetitive and consummatory processes in rats

Keshav S. Subramanian [1,2], Logan Tierno Lauer[1], Anna M. R. Hayes [1], Léa Décarie-Spain[1], Kara McBurnett[1], Anna C. Nourbash[1], Kristen N. Donohue[1], Alicia E. Kao[1], Alexander G. Bashaw[1,2], Denis Burdakov [3], Emily E. Noble [4], Lindsey A. Schier[1,2] & Scott E. Kanoski [1,2] ✉

The lateral hypothalamic area (LHA) integrates homeostatic processes and reward-motivated behaviors. Here we show that LHA neurons that produce melanin-concentrating hormone (MCH) are dynamically responsive to both food-directed appetitive and consummatory processes in male rats. Specifically, results reveal that MCH neuron $Ca^{2+}$ activity increases in response to both discrete and contextual food-predictive cues and is correlated with food-motivated responses. MCH neuron activity also increases during eating, and this response is highly predictive of caloric consumption and declines throughout a meal, thus supporting a role for MCH neurons in the positive feedback consummatory process known as appetition. These physiological MCH neural responses are functionally relevant as chemogenetic MCH neuron activation promotes appetitive behavioral responses to food-predictive cues and increases meal size. Finally, MCH neuron activation enhances preference for a noncaloric flavor paired with intragastric glucose. Collectively, these data identify a hypothalamic neural population that orchestrates both food-motivated appetitive and intake-promoting consummatory processes.

Caloric regulation is determined by the integration of appetitive food-motivated responses and consummatory physiological processes that govern meal size[1–3]. Once eating is initiated, the amount of calories consumed during a meal is regulated by two opposing processes: an early meal positive feedback process known as appetition that promotes further consumption[3,4], and a later meal negative feedback process known as satiation that leads to meal termination[1]. While the neurobiological systems that regulate premeal appetitive responses and meal-termination satiation signaling have been widely investigated, the neural substrates mediating appetition remain elusive. Here we seek to identify how the brain integrates appetitive and consummatory signals, specifically those driving appetition, to promote caloric consumption.

The hypothalamus regulates both appetitive and consummatory behaviors. Regarding appetitive food-motivated behaviors, Agouti-related Protein (AgRP)-expressing neurons in the arcuate nucleus of the hypothalamus (ARH) have extensively been shown to potently and reliably trigger food-seeking responses[5–8]. However, given that

[1]Human and Evolutionary Biology Section, Department of Biological Sciences, Dornsife College of Letters, Arts and Sciences, University of Southern California, Los Angeles, California, USA. [2]Neuroscience Graduate Program, University of Southern California, Los Angeles, California, USA. [3]Department of Health Sciences and Technology, ETH Zurich, Zurich, Switzerland. [4]Department of Nutritional Sciences, University of Georgia, Athens, USA. ✉e-mail: kanoski@usc.edu

fasting-induced AgRP neuron activity is inhibited upon access to food consumption[9] or exposure to food-associated cues[7], these neurons are unlikely to serve as key integrators of appetitive and consummatory processes. Orexin (aka, hypercretin)-producing neurons in the lateral hypothalamic area (LHA) are also associated with foraging and other premeal appetitive responses[10–12]. However, like AgRP neurons, orexin neuron activity ceases immediately upon eating[13,14], and therefore, similar to AgRP neurons, orexin neurons are also unlikely candidates to integrate appetitive processes with prandial appetition. Melanin-concentrating hormone (MCH)-producing neurons, also located in the LHA but distinct from orexin neurons, are glucose-responsive[15,16] and pharmacological MCH administration increases food intake via an increase in meal size[17], thus making them a feasible candidate population of neurons for such integration. Furthermore, the MCH receptor, MCH-1R, is required for palatable food-associated cues to promote overeating in mice[18], thus supporting a role for MCH signaling in linking conditioned appetitive behaviors with consummatory intake-promoting processes.

Here we aim to understand the role of MCH neurons in integrating food-motivated appetitive and intake-promoting consummatory processes. Our findings reveal that physiological MCH neuron Ca$^{2+}$ activity increases upon exposure to both discrete and contextual-based food-predictive cues and that these MCH neural responses are strongly associated with appetitive cue-induced behavioral actions. In addition to contributing to appetitive processes, MCH neuron Ca$^{2+}$ activity dynamically increases during eating, and this response is positively correlated with calories consumed during a meal. This eating-induced elevation of MCH neuron activity also dampens throughout the course of a meal, which supports the involvement of these neurons in promoting early-phase eating (appetition). Lastly, chemogenetic results functionally validate the physiological MCH neural responses, as MCH neuron activation increased appetitive responses to food-predictive cues, elevated meal size under the same conditions as the Ca$^{2+}$ imaging consummatory analyses, and enhanced flavor-nutrient learning. Taken together, these findings support a role for MCH neurons in integrating appetite and appetition.

## Results

### MCH neuron Ca$^{2+}$ activity increases during discrete sucrose-predictive Pavlovian cues and is associated with appetitive responses

To determine if MCH neurons are responsive to discrete exteroceptive food-predictive cues, rats were injected with an AAV9.pMCH.G-CaMP6s.hGH (MCH promoter-driven GCaMP6s) followed by an optic fiber implanted into the LHA (Fig. 1a). To confirm the selectivity of the MCH promoter, immunofluorescence colocalization analyses reveal that the GCaMP6s fluorescence signal was exclusive to MCH immunoreactive neurons (Fig. 1b). Results from MCH GCAMP6s neuroanatomical quantification show that using this approach approximately 70.9 ± 3.5% (standard error of the mean) of LHA MCH neurons and 18.9 ± 1.1% of perifornical area MCH neurons were colocalized with the MCH GCAMP6s. Post viral transduction, the animals were trained to associate one auditory cue with access to sucrose (conditioned-stimulus positive, CS+) and a different auditory cue with no sugar access (CS−). Then, fiber photometry was used to measure MCH neuron Ca$^{2+}$ activity in response to the CS+ and CS− (Fig. 1c). The animals readily learned the Pavlovian discrimination, exhibited a significantly increased number of licks per trial, reduced latency to lick, and increased number of CS+ trials with a consummatory response at the end (Day 7) vs. beginning (Day 1) of training (Fig. 1d–f). During the test recording session on Day 7 of training, physiological MCH neuron Ca$^{2+}$ activity reflected the learned CS discrimination, such that there was increased activity during the CS+ presentation in comparison to the CS− (Fig. 1g–i). Moreover, this effect was specific to the CS period, as there were no differences 5s-pre-CS and 5s-post-CS between the CS+

and CS− (Fig. 1j). Additionally, MCH neuron Ca$^{2+}$ activity was negatively correlated with latency to lick, such that the CS+ induced MCH neuron Ca$^{2+}$ response was predictive of a faster appetitive response to initiate sucrose consumption upon cue offset (Fig. 1k). Finally, that food cue-induced MCH neuron responses reflect learning is further supported by findings that elevated CS+ induced Ca$^{2+}$ responses are observed on Day 5 of training, corresponding with statistical evidence of learning in behavioral measures, but not on Day 2 of training where behavioral analyses are subthreshold to support learning (Supplementary Fig. 6). These data indicate that physiological MCH neuron Ca$^{2+}$ activity increases to and is associated with behavioral appetitive responsivity to discrete food-predictive cues.

### MCH neuron Ca$^{2+}$ activity increases in response to contextual-based food-predictive cues and is associated with food-seeking behavior

To assess whether physiological MCH neuron Ca$^{2+}$ activity is engaged by contextual cues associated with food, rats with the MCH promoter-driven GCaMP6s in the LHA were trained and tested in a palatable food-reinforced conditioned place preference (CPP) procedure in which one context is associated with access to highly palatable food and another is not (Fig. 2a, left)[19–21]. To evaluate MCH neuron activity based on learned food-associated contextual cues independent of food consumption, on the test day, the animals had access to both food-paired and unpaired contexts while food was not present, and MCH neuron Ca$^{2+}$ activity was measured via a fiber photometry system (Fig. 2a, right). Results reveal that animals successfully learned to prefer the food-paired context as exhibited by an increased percentage of time spent on the paired context during testing and the preference shift from baseline (difference in percentage time spent on side pre- and post-training) relative to the unpaired context (Fig. 2b–d). During testing, MCH neuron Ca$^{2+}$ activity was elevated when the animals were on the food-paired context compared to the unpaired context throughout the test session (Fig. 2e). However, the magnitude of this response was not correlated with contextual cue-based food-seeking behavior (assessed as preference shift from baseline for the paired context; Fig. 2f). MCH neuron Ca$^{2+}$ activity was also increased upon the first 2-s entry period into the paired compared to the unpaired context (Fig. 2g). In this case, the magnitude of the MCH neuron Ca$^{2+}$ response upon entry to the paired context was positively associated with contextual-based food-seeking behavior (Fig. 2h). These data indicate that physiological MCH neuron Ca$^{2+}$ activity increases upon entry into a context associated with palatable food access, and that this response is linked to food-seeking memory based on contextual cues under test conditions where food was not available.

### MCH neuron Ca$^{2+}$ activity increases during eating and is highly predictive of eating bout duration and caloric intake during a meal

To evaluate whether physiological MCH neuron Ca$^{2+}$ activity is altered during consummatory processes, rats with the MCH promoter-driven GCaMP6s in the LHA were allowed to refeed on chow following an overnight fast and MCH neuron Ca$^{2+}$ activity was recorded and time-stamped to active eating behavior and interbout intervals when animals were not eating (Fig. 3a, b). Results reveal that MCH neuron Ca$^{2+}$ activity dynamically increased during eating in comparison to interbout periods and that this effect is greatest during the early part of the meal (Fig. 3c–e; representative trace from a single animal in 3c). In addition, MCH neuron Ca$^{2+}$ activity was significantly elevated 5-min post voluntary meal termination as compared to the 5-min pre-food access period (Fig. 3e). Combined, these data indicate that physiological MCH neuron Ca$^{2+}$ activity dynamically increases during active

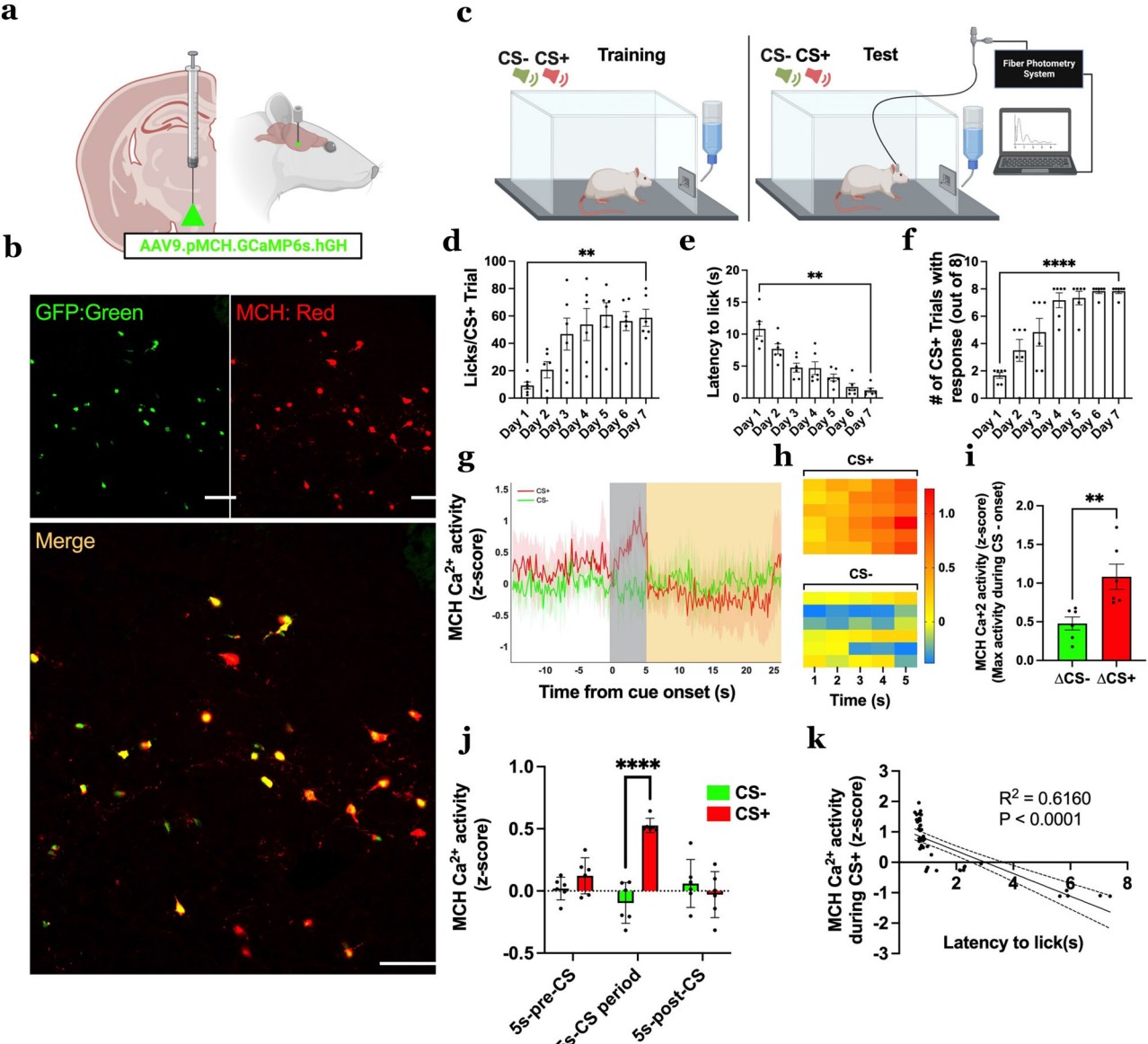

**Fig. 1 | Physiological MCH neuron Ca²⁺ activity increases in response to discrete food-predictive cues and is associated with behavioral appetitive responsivity.**
**a** Schematic diagram depicting a viral approach to record physiological MCH neuron Ca²⁺ activity with fiber photometry in rats ($n = 6$). An adeno-associated virus containing an MCH promoter-driven GCaMP6s (AAV9.pMCH.GCaMP6s.hGH) is injected into the LHA and an optic fiber was implanted above the injection site.
**b** Representative images of fluorescent reporter in MCH GCaMP6s colocalizing with MCH immunoreactive neurons (repeated and verified independently in $n = 6$ rats). **c** Schematic cartoon depicting fiber photometry recording of MCH neuron Ca²⁺ activity during the Pavlovian discrimination task ($n = 8$ CS+, $n = 8$ CS− cues). **d**–**f** Training data for Pavlovian discrimination task (data were analyzed using a one-way ANOVA with repeated measures and multiple comparisons, $n = 6$ rats): **d** Average number of licks for sucrose solution per CS+ trial, **e** Average latency to lick from sucrose solution per CS+ trial, and **f** Average number of CS+ trials with a response via licking sucrose solution. **g**–**k** Fiber photometry recording of MCH neuron Ca²⁺ activity during the test phase (data analyzed using Student's two-tailed paired $t$-test, $n = 6$ rats, with **g** MCH neuron Ca²⁺ activity (z-score) time-locked to cue onset (CS+ in red and CS− in green; −15 to 25 s relative to the start of the 5 s cue [gray box]). **h** Heatmap of the MCH neuron Ca²⁺ activity (z-score) for each animal during each of the 5 s of CS presentation. **i** MCH neuron Ca²⁺ activity during cue period [gray box] (max activity during CS − activity during cue onset), **P = 0.0042, **j** MCH neuron Ca²⁺ activity for 5s-pre-CS, 5 s CS period and 5s-post-CS, compared between CS, **P = 0.001 and **k** Simple linear regression of MCH neuron Ca²⁺ activity (max activity during CS − activity during cue onset) during each CS+ trial relative to latency to lick from sucrose solution. The solid line is linear fit to data and dashed lines are 95% confidence interval error bars, $R^2 = 0.5039$, $P = 1.08E-10$. Data shown as mean ± SEM; Scale = 100 μm; **P < 0.01, ****P < 0.0001. Source data are provided as a Source Data file. Created with Biorender.com.

eating behavior and that MCH neuron Ca²⁺ activity tone is higher in the satiated vs. fasted state. Additional analyses show that average ΔMCH neuron Ca²⁺ activity within an eating bout and overall meal ΔAUC (AUC 5-min post-last bout − AUC 5-min pre-food access) were both strongly correlated with cumulative chow intake, indicating that physiological MCH neuron Ca²⁺ activity is highly predictive of caloric intake within a meal (Fig. 3f, g). The ΔMCH neuron Ca²⁺ activity within an eating bout negatively correlated with the temporal phase of the meal [bouts occurring in the first (early), second (mid), or third (late) tertile of the meal] such that MCH neuron Ca²⁺ activity was greatest during the first part of the meal, and waned as the rats approached meal termination (Fig. 3h). The ΔMCH neuron Ca²⁺ activity was also positively correlated with eating bout duration (Fig. 3i). These latter two correlations suggest that MCH neuron responses are tightly coupled to ingestive behaviors subserving appetition (e.g., rapid invigoration of consummatory actions and sustained periods of eating). Overall these data

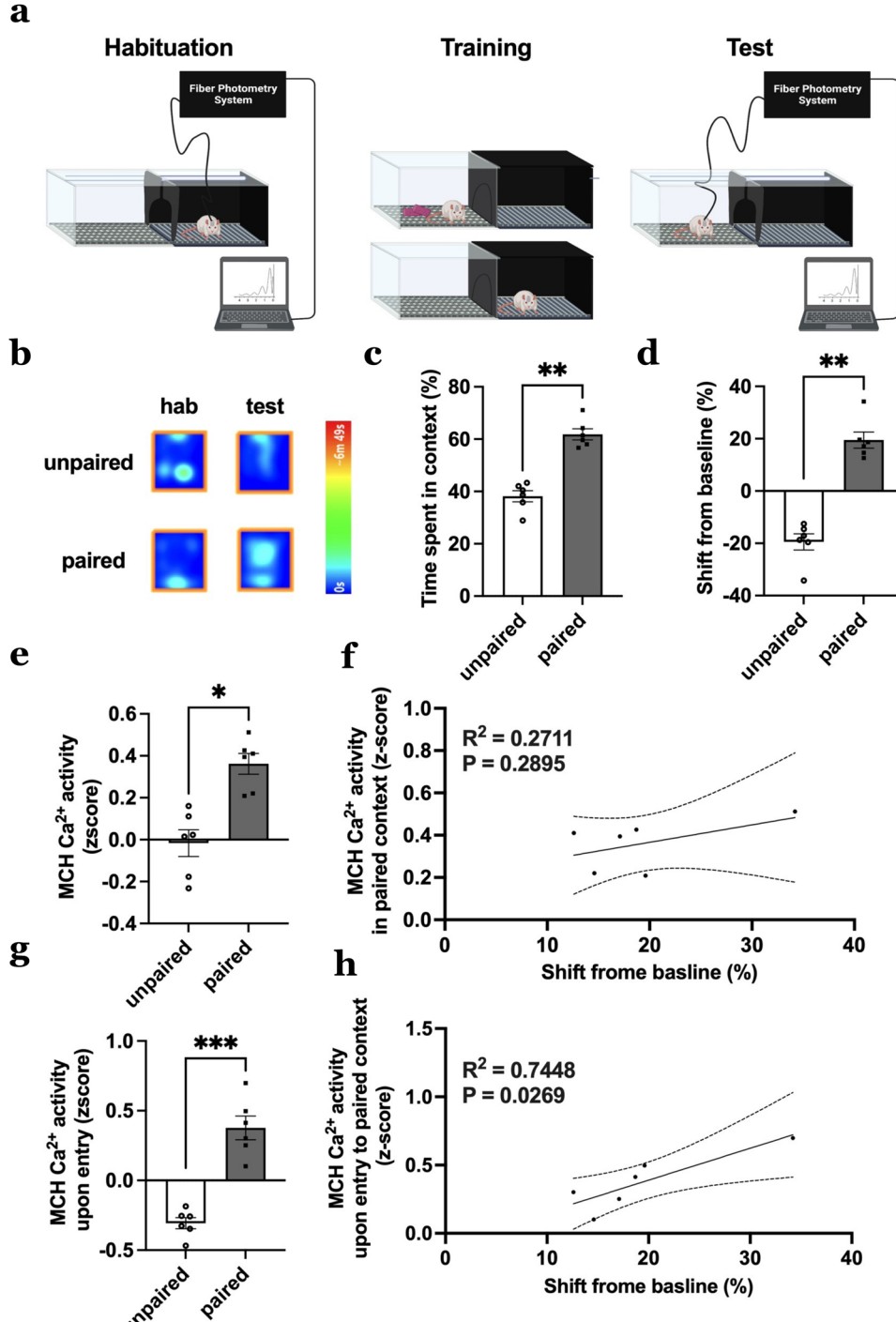

**Fig. 2 | Physiological MCH neuron Ca²⁺ activity increases in response to contextual-based food-predictive cues and is associated with food-seeking memory behavior. a** Schematic diagram depicting fiber photometry recording of MCH neuron $Ca^{2+}$ activity during conditioned place preference (CPP) in rats ($n = 6$). **b** Representative heat maps (single animal) depicting the time spent on the food-paired or unpaired side during habituation and test day. The less preferred side during habituation is assigned as the food-paired side during training. **c, d** CPP behavior data during the test phase (data analyzed using Student's two-tailed paired $t$-test, $n = 6$ rats), with **c** percentage time spent on a side, **\*\*$P = 0.0027$** and **d** shift from baseline (difference in percentage time spent on the side between pre- and post-training), **\*\*$P = 0.0016$**. **e–h** Fiber photometry recording of MCH neuron

$Ca^{2+}$ activity during CPP test phase (data analyzed using Student's two-tailed paired $t$-test, $n = 6$ rats) with **e** MCH neuron $Ca^{2+}$ activity on a side, **\*$P = 0.0133$**, **f** simple linear regression of MCH neuron $Ca^{2+}$ activity on food-paired side relative to shift from baseline $R^2 = 0.2711$, $P = 0.2895$, **g** MCH neuron $Ca^{2+}$ activity upon entry into a side, **\*\*\*$P = 0.0001$** and **h** simple linear regression of MCH neuron $Ca^{2+}$ activity upon entry into food-paired side relative to shift from baseline, $R^2 = 0.7448$, $P = 0.0269$. For all linear regression analysis, the solid line is linear fit to data and dashed lines are 95% confidence interval error bars. Data shown as mean ± SEM; *$P < 0.05$, **$P < 0.01$, ***$P < 0.001$. Source data are provided as a Source Data file. Created with Biorender.com.

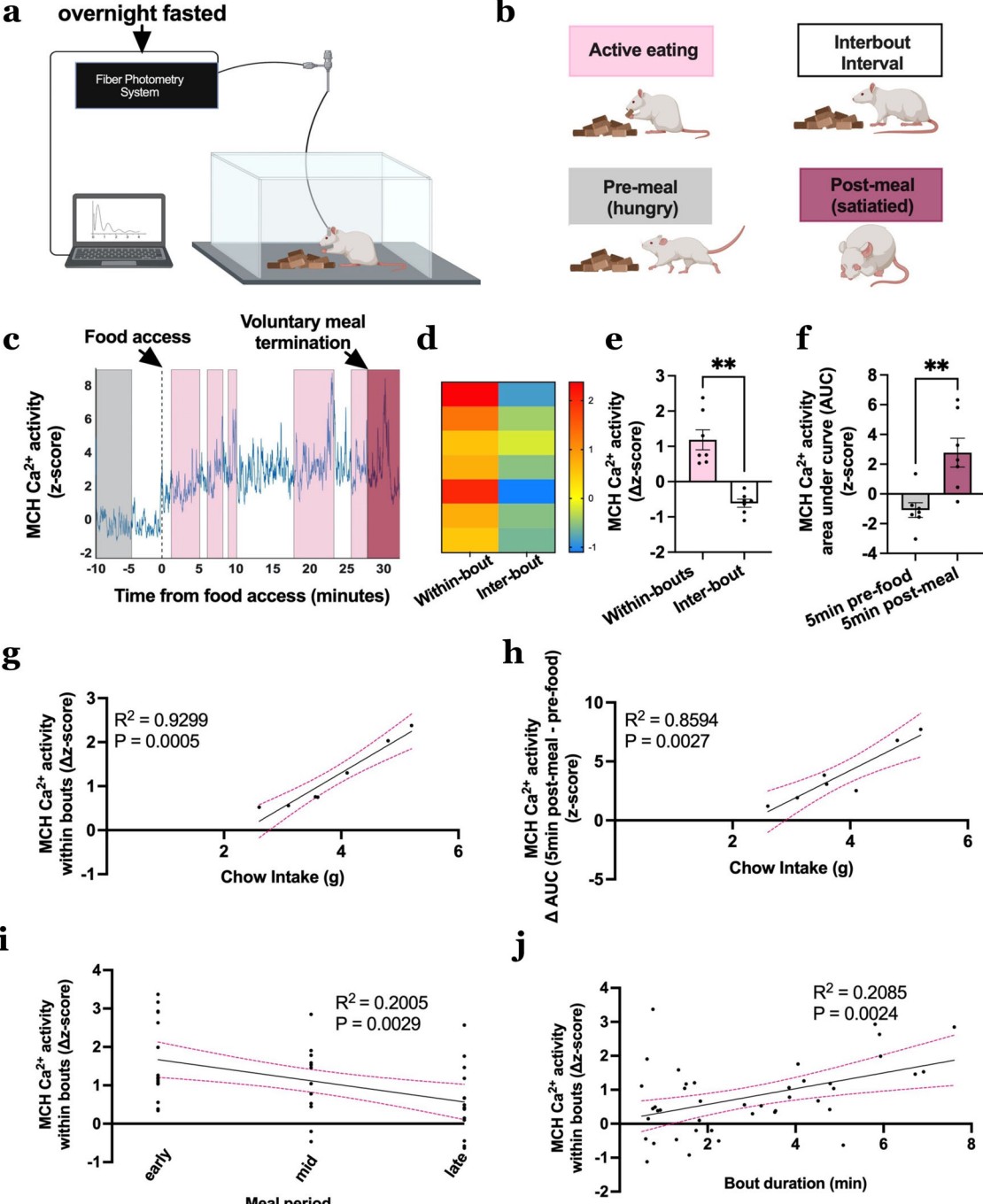

**Fig. 3 | Physiological MCH neuron Ca²⁺ activity increases during active feeding and is highly predictive of cumulative caloric intake and eating bout duration within a meal. a** Schematic diagram depicting fiber photometry recording of MCH neuron Ca²⁺ activity during refeeding after an overnight fast in rats ($n = 7$).
**b** Schematic cartoon depicting different feeding phases of refeeding after the fast.
**c** Representative trace of MCH neuron Ca²⁺ activity during refeeding after a fast time locked to food access (−10 to 30 min relative to the start food access [dotted vertical line]). Premeal [gray box], active eating [pink boxes; quantified as the difference in activity between start and end of eating bout], interbout intervals [white boxes; quantified as the difference in activity between after ending eating bout and before starting next eating bout] and voluntary satiation [dark pink box] are represented. **d** Heatmap representing each animal's MCH neuron Ca²⁺ signal during active eating bouts and interbout intervals **e**–**j** Fiber photometry recording of MCH neuron Ca²⁺ activity during refeeding after a fast (data analyzed using Student's two-tailed paired *t*-test, $n = 7$ rats) with **e** MCH neuron Ca²⁺ activity within eating bouts [pink box] and interbout intervals [white box], **P = 0.003, **f** Area under curve (AUC) MCH neuron Ca²⁺ activity during 5 min pre-food access [gray box] and 5 min post-last bout [dark pink box], **P = 0.0061, **g** Simple linear regression of MCH neuron Ca²⁺ activity within eating bouts relative to cumulative chow intake, $R^2 = 0.9299$, ***P = 0.0005, and **h** Simple linear regression of ΔAUC (AUC post-last-bout − AUC pre-food access) MCH neuron Ca²⁺ activity relative to cumulative chow intake, $R^2 = 0.8594$, **P = 0.0027, **i** Simple linear regression of MCH neuron Ca²⁺ activity within eating bouts relative to different time points in meal period (early: first tertile, mid: second tertile, late: last tertile), $R^2 = 0.2005$, **P = 0.0029 and **j** Simple linear regression of MCH neuron Ca²⁺ activity within eating bouts relative to bout duration, $R^2 = 0.2085$, **P = 0.0024. For all linear regression analyses, the solid line is linear fit to data, and the dashed lines are 95% confidence interval error bars. Data are shown as mean ± SEM; **P < 0.001. Source data are provided as a Source Data file. Created with Biorender.com.

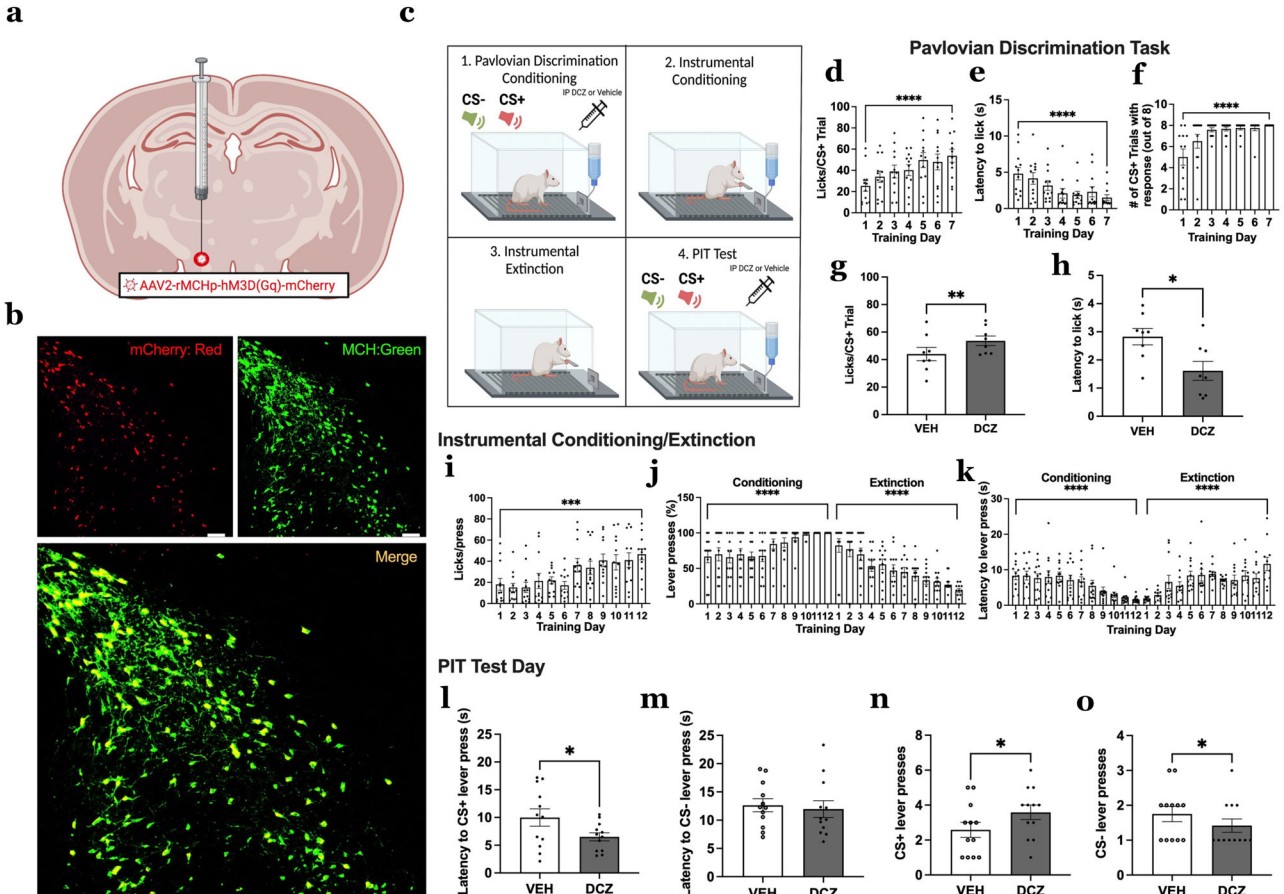

**Fig. 4 | MCH neuron activation increases appetitive responses to discrete food-predictive cues. a** Schematic diagram depicting a viral approach to chemogenetically activate MCH neurons. An adeno-associated-virus containing excitatory MCH DREADDs-mCherry transgene (AAV2-rMCHp-hM3D(Gq)-mCherry) is injected into the LHA/ZI. **b** Representative images of fluorescent reporter in MCH DREADDs colocalizing with MCH immunoreactive neurons (repeated and verified independently in $n = 8$ rats) **c** Schematic cartoon depicting chemogenetic activation of MCH neurons during Pavlovian-Instrumental-Transfer (PIT). **d–f** Training data for Pavlovian discrimination task (data were analyzed using a one-way ANOVA with repeated measures and multiple comparisons, $n = 8$ rats): **d** Average number of licks for sucrose solution per CS+ trial, **e** Average latency to lick from sucrose solution per CS+ trial and **f** average number of CS+ trials with a response via licking sucrose solution. **g, h** Effects of chemogenetic activation of MCH neurons during the test phase of Pavlovian discrimination task in rats (data analyzed using

Student's two-tailed paired $t$-test, $n = 8$ rats) with **g** average number of licks for sucrose solution per CS+ trial, **\*\***$P = 0.0022$ and **h** average latency to lick from sucrose solution per CS+ trial, **\***$P = 0.0287$. **i–k** Training data for instrumental conditioning/extinction (data were analyzed using a one-way ANOVA with repeated measures and multiple comparisons, $n = 12$ rats): **i** Average number of licks for sucrose solution per lever press during conditioning, **j** Percentage number of lever presses represented for conditioning and extinction and **k** Latency to lever press for conditioning and extinction. **l–o** Effects of chemogenetic activation of MCH neurons during test phase for PIT in rats (data analyzed using Student's two-tailed paired $t$-test, $n = 12$ rats) with **l** latency to lever press after the CS+, **\***$P = 0.0498$, **m** and CS− cue, $P = 0.6894$, **n** Number of lever presses after the CS+, **\***$P = 0.0323$ **o** and CS− cue, **\***$P = 0.0388$. Data shown as mean ± SEM; Scale = 100 μm; **\***$P < 0.05$, **\*\***$P < 0.01$, **\*\*\***$P < 0.001$, **\*\*\*\***$P < 0.0001$. Source data are provided as a Source Data file. Created with Biorender.com.

---

indicate that physiological MCH neuron Ca²⁺ activity is tightly responsive to early meal consummatory behaviors and predictive of total caloric intake within a meal.

### Activation of MCH neurons increases appetitive responses to discrete sucrose-predictive Pavlovian cues

To confirm that the elevated MCH neuron Ca²⁺ activity in response to food-predictive cues is functionally relevant to cue-induced appetitive behavior, we utilized a virogenetic approach to chemogenetically activate MCH neurons. Rats were injected with an AAV2-MCH DREADDs-hM3D(Gq)-mCherry (MCH DREADDs) targeting the LHA and zona incerta (ZI) with excitatory DREADDs (designer receptors exclusively activated by designer drugs) under the control of an MCH promoter (Fig. 4a). Previous work from our lab has shown that this approach transfects approximately ~80% of all MCH neurons and is highly selective to MCH neurons[22], which is consistent with the immunofluorescence histological chemistry results in the present

study (Fig. 4b). Animals were trained in the Pavlovian-Instrumental-Transfer (PIT) task, which measures instrumental appetitive actions in response to discrete food-predictive cues under conditions where food is not available[23,24]. PIT consists of four phases: (1) Pavlovian discrimination conditioning, (2) instrumental conditioning, (3) instrumental extinction, and (4) PIT test day. On test day, animals received intraperitoneal (IP) injections of Deschloroclozapine (DCZ; DREADDs ligand) or vehicle (1% DMSO in 99% saline) to selectively increase MCH neuron activity (Fig. 4c). Results reveal that animals successfully learned the Pavlovian discrimination as exhibited by increased licking, reduced latency to lick, and increased percentage of CS+ trials with an appetitive response across training (Fig. 4d–f). Further, DCZ-induced activation of MCH neurons during the Pavlovian discrimination phase led to an increased number of licks and reduced latency to lick per CS+ trial (Fig. 4g, h), thus revealing that the physiological MCH neuron Ca²⁺ responses to the CS+ during Pavlovian conditioning are functionally relevant to cue-induced appetitive behavior. Importantly, IP DCZ had

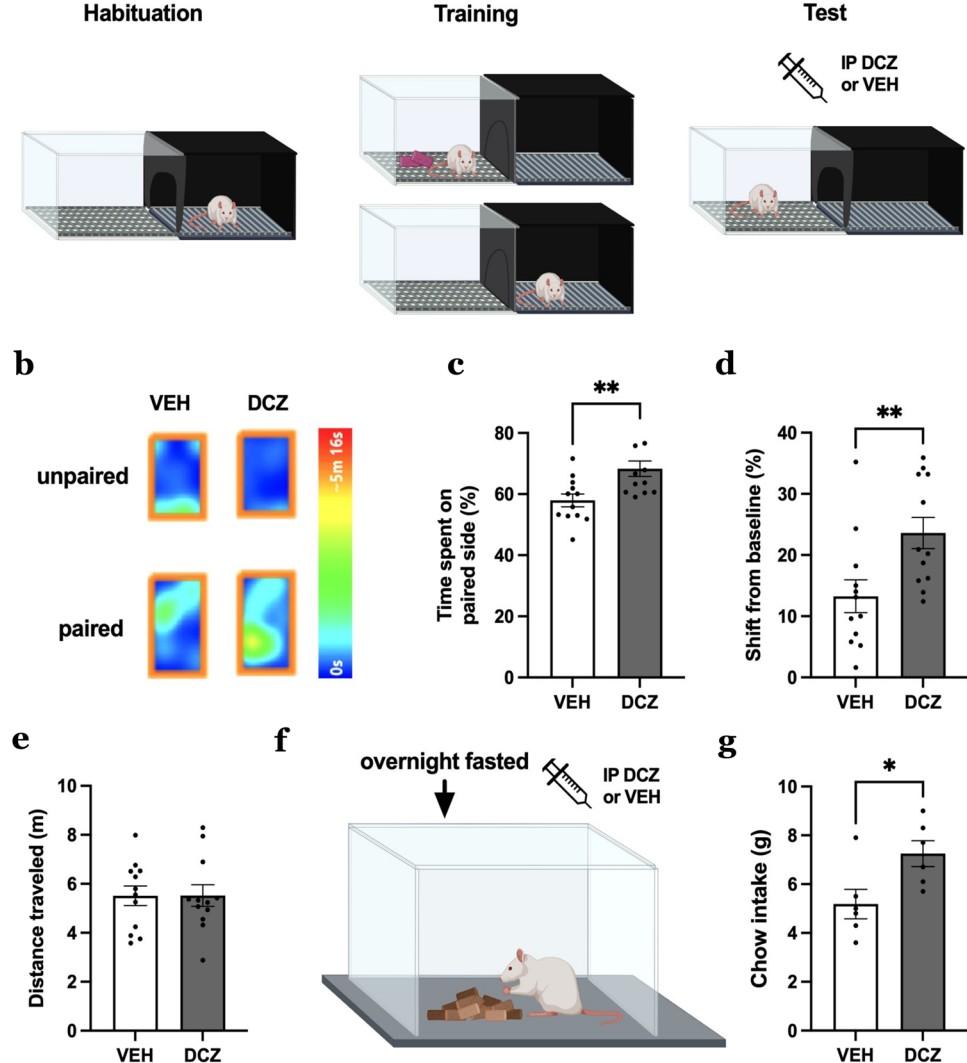

**Fig. 5 | MCH neuron activation increases appetitive responses to contextual-based food-predictive cues and meal size after an overnight fast. a** Schematic diagram of chemogenetic activation of MCH neurons during CPP in rats ($n = 12$). **b** Representative heat maps (single animal) depicting the time spent on the food-paired or unpaired side between vehicle and DCZ treatments. **c**–**e** Effects of chemogenetic activation of MCH neurons during CPP test phase (data analyzed using Student's two-tailed paired $t$-test, $n = 12$ rats) with **c** percentage time spent on the food-paired side,**P = 0.0028** and **d** shift from baseline (difference in percentage time spent on the side between pre- and post-training), **P = 0.0028. e** Total distance traveled during CPP test day ($P = 0.9829$). **f, g** Effects of chemogenetic activation of MCH neurons during refeeding after a fast in rats (data analyzed using Students two-tailed test, $n = 12$ rats) with **f** schematic diagram of chemogenetic activation of MCH neurons during refeeding after a fast and **g** cumulative chow intake after refeeding after a fast, *P = 0.0277. Data shown as mean ± SEM; *P < 0.05, **P < 0.01. Source data are provided as a Source Data file. Created with Biorender.com.

no impact on appetitive responsivity in the Pavlovian discrimination task without the MCH DREADDs expressed in MCH neurons (control MCH promoter AAV; Supplementary Fig. 1). Throughout phase 2 (instrumental conditioning), animals showed increased licking, reduced latency to lick, and increased the number of lever presses for the sucrose solution across training (Fig. 4j, k). In phase 3 (instrumental extinction), animals increased their latency to lever press and overall reduced the number of lever presses per session, suggesting that they learned that lever pressing no longer provides reinforcement (Fig. 4j, k). In phase 4 (PIT test day), chemogenetic activation of MCH neurons reduced latency to lever press when the CS+ was presented, yet had no effect on latency following CS− presentation (Fig. 4l, m). Furthermore, MCH neuron activation increased the total number of lever presses following the CS+ presentation, but decreased the number of presses following the CS− presentation (Fig. 4n, o). Overall, these data show that chemogenetic activation of MCH neurons increases appetitive behavioral responses to food-predictive cues.

## MCH neuron activation increases food seeking based on contextual food cues

To determine whether MCH neuron Ca$^{2+}$ responses reflect a functional role for MCH neurons in driving food seeking based on contextual cues, animals with the MCH promoter-driven excitatory DREADDs underwent CPP training and testing (Fig. 5a). Results reveal that chemogenetic activation of MCH neurons during CPP testing increased the percentage of time spent on the food-paired context and increased the shift from baseline for the paired context (difference in time spent on side pre- and post-training), indicating that activation of MCH neurons increases appetitive responses to contextual-based food-predictive cues (Fig. 5b–d). There were no differences in total distance traveled on CPP test day (Fig. 5e), suggesting that MCH neuron activation did not influence locomotor activity. IP DCZ had no impact on appetitive responsivity in CPP in animals without the MCH DREADDs expressed in MCH neurons (control MCH promoter AAV; Supplementary Fig. 3).

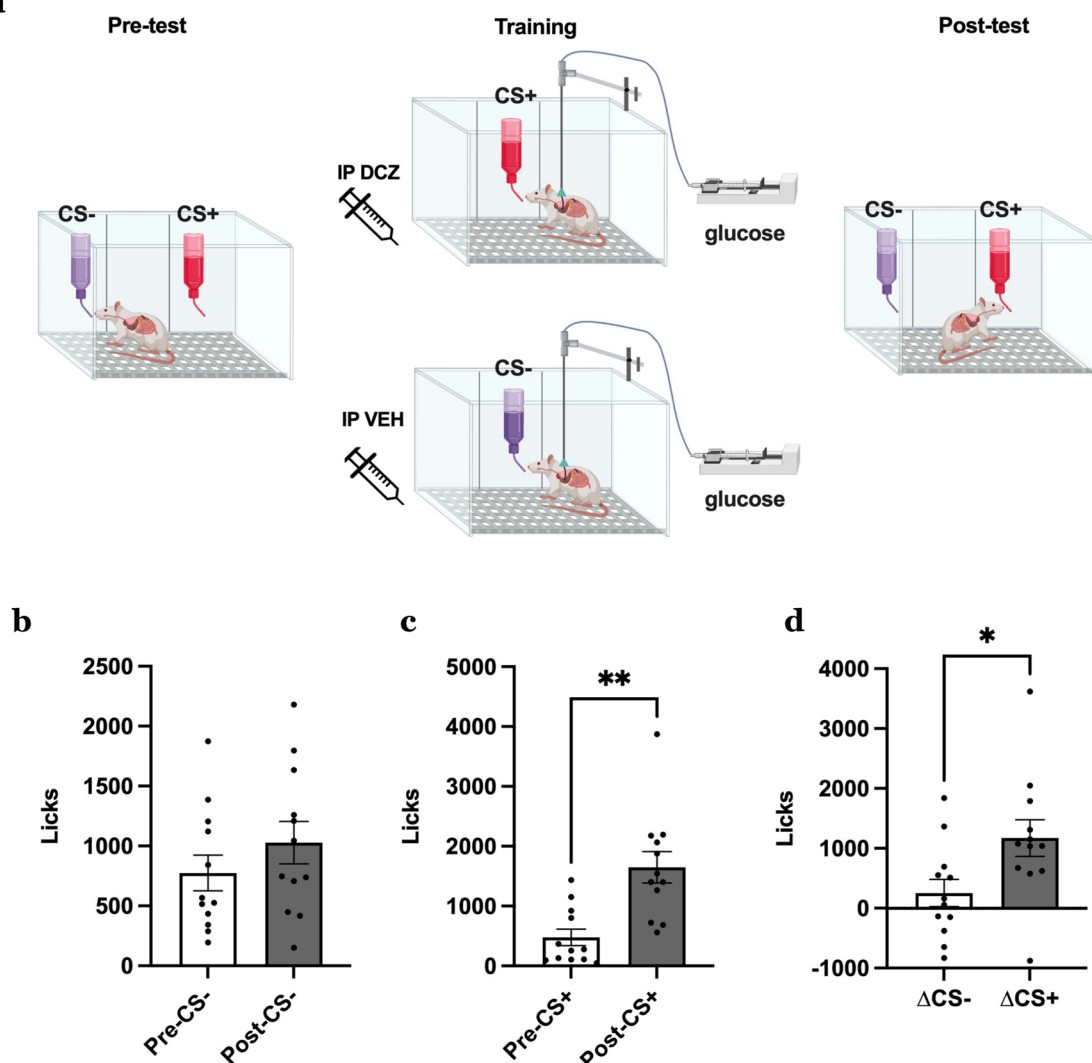

**Fig. 6 | Activation of MCH neurons enhances preference for a noncaloric flavor paired with intragastric (IG) glucose. a** Schematic diagram depicting the paradigm for chemogenetic activation of MCH neurons during flavor preference conditioning paired with IG glucose in rats ($n = 12$). **b–d** Effects of chemogenetic activation of MCH neurons paired with IG glucose (data analyzed using Student's two-tailed paired $t$-test, $n = 12$ rats) with **b** number of licks for the vehicle-paired CS−, **c** and DCZ-paired CS+ flavors during pre- and post-tests, **\*\***$P = 0.0028$ and **d** difference in the number of licks for CS between pre- and post-tests, **\***$P = 0.0353$. Data shown as mean ± SEM; \*$P < 0.05$, \*\*$P < 0.01$. Source data are provided as a Source Data file. Created with Biorender.com.

### MCH neuron activation increases meal size after a fast

To examine whether MCH neuron Ca$^{2+}$ responses during eating are functionally relevant to cumulative caloric consumption, animals with excitatory MCH promoter DREADDS were tested under the same eating conditions as the previous Ca$^{2+}$ imaging experiment, where ad libitum chow was offered following a 24-h fast and animals consumed food until voluntary meal termination. Results reveal that chemogenetic activation of MCH neurons increased meal size under these conditions (Fig. 5f). In addition, chemogenetic activation of MCH neurons via IP DCZ increased home cage chow intake under ad libitum free-feeding conditions, which is consistent with our previous work using a different DREADDs ligand, Clozapine-N-oxide[22]. IP DCZ had no impact, however, on home cage intake in animals without the MCH DREADDS expressed in MCH neurons (control MCH promoter AAV; Supplementary Fig. 4).

### MCH neuron activation promotes a preference for noncaloric flavor paired with IG glucose

In addition to driving appetition within a meal, positive interoceptive events (e.g., nutrition) condition lasting preferences for the associated flavor[3,4]. Thus, here we evaluate whether MCH neuron activation bolsters this type of flavor-nutrient learning. Animals that were surgically implanted with a gastric catheter and had excitatory MCH promoter DREADDs went through training to associate two noncaloric saccharin-sweetened flavors (CS+) with intragastric glucose infusion. One flavor was paired with DCZ injection-induced MCH neuron activation (CS+) and the other with vehicle injections (CS−). Preference for the two flavors, where animals were given access to both flavors without drug treatments or infusions, was measured both before and after training, (Fig. 6a). Results reveal increased consumption of the CS+ post-training relative to pre-training, whereas there were no pre- vs. post-training differences for the CS− (Fig. 6b, c). In addition, the difference in the number of licks pre- and post-training was increased for the CS+ relative to the CS− (Fig. 6d). These findings indicate that MCH neuron activation enhanced flavor-nutrient conditioning. IP DCZ did not, however, induce flavor-nutrient preference conditioning without the MCH DREADDs expressed in MCH neurons (control MCH promoter AAV; Supplementary Fig. 5).

## Discussion

Once eating is initiated, hedonic orosensory gustatory processes are thought to be the main driving factors in promoting further food consumption, whereas post-oral gut-mediated processes are functionally linked with satiating processes leading to meal termination. However, work from Sclafani and colleagues using flavor-nutrient conditioning procedures elucidated an early meal positive feedback process known as appetition, which acts through post-oral processes to sustain ongoing consummatory behaviors[4]. The neural substrates underlying appetition, however, remain elusive. Here we show evidence that LHA MCH-producing neurons participate in the appetition process. Specifically, MCH neurons dynamically respond to both appetitive and consummatory signals. That these physiological neural responses are involved in mediating appetition is supported by our results revealing that MCH neuron activity increases during eating and is strongly predictive of cumulative caloric intake during a meal, eating bout duration, and, importantly, is augmented during eating bouts that occur earlier vs. later in the course of a meal. These latter findings suggest that MCH neurons function to prolong eating bout duration, particularly during the early prandial stage, and that these responses contribute to overall larger meal consumption.

The involvement of MCH neurons in promoting appetition has been previously proposed[25,26] and is also supported by a recent study revealing that optogenetic activation of MCH neurons during consumption increases food intake, and this effect was specific to time-locking MCH neuron activation to active eating periods[27]. Additionally, optogenetic activation of MCH neurons in mice reversed the preference for sucrose over non-caloric sucralose[28], indicating that MCH neurons may provide a nutritive signal even in the absence of calories. Here we reasoned that if MCH neurons are involved in appetition, then their activity should be associated with enhanced flavor-nutrient preference conditioning, potentially by augmenting the reinforcing properties of post-oral nutrient processing. Indeed, MCH neuron activation promoted a preference for a noncaloric saccharin-sweetened flavored solution paired with intragastric (IG) glucose infusion. However, it is important to note that these results may be mediated by MCH neuron activity enhancing hedonic taste-mediated processes, post-oral nutritive processes (i.e., appetition), or both. Indeed, pharmacological MCH enhances positive orofacial responses to sucrose during a taste reactivity test via downstream opioid signaling[29], which suggests that MCH is interacting with the opioid system to enhance the hedonic flavor processes. An interesting follow-up direction would therefore be to record physiological MCH neuron Ca²⁺ activity while animals have IG glucose and fructose infusions, as such an experiment bypasses orosensory processes, and post-oral glucose has been shown to promote flavor-nutrient preference conditioning whereas fructose does not[30]. It is also important to note that MCH receptor knockout mice have intact post-ingestive glucose-mediated flavor-nutrient conditioning[31]. While this might suggest that MCH is not involved with appetition processing, this could be the result of putative compensatory mechanisms based on genetic developmental MCH receptor defects. Further, MCH neurons express transcripts for multiple neuropeptides and neurotransmitter markers[32], and thus it is possible that activation of MCH receptors is not sufficient to drive MCH neuron-mediated effects on appetition.

In addition to associations with consummatory processes, our results reveal that physiological MCH neuron Ca²⁺ activity increases in response to discrete and contextual-based food-predictive cues, and that chemogenetic MCH neuron activation increases appetitive food-seeking behaviors. These results cannot be secondary to consumption as MCH Ca²⁺ responses during CPP testing were evaluated without food present during the test days, and further, MCH neuron-induced operant responses during PIT testing were not reinforced with food. It is also unlikely that MCH neuron-associated appetitive responses were secondary to effects on general locomotor activity, as chemogenetic activation of MCH neurons during PIT resulted in fewer lever presses following CS− presentations and did not influence locomotor activity during CPP testing, and moreover, previous work has shown that MCH neuron activation does not impact short-term locomotor activity[27]. Altogether, these findings suggest that MCH neuron-associated appetitive responses are not secondary to food consumption or altered physical activity, but, more likely, are neural and behavioral responses induced by food-predictive cues.

It is unclear how MCH neurons are signaling to integrate appetitive and consummatory behaviors. A possible mechanism is that increased MCH neuron Ca²⁺ activity during appetitive behaviors enhances the mental imagery of food in response to food-associated cues, resulting in the animals showing stronger appetitive drive. In regard to consummatory behaviors, MCH signaling may be augmenting either the oral sensory component of eating (hedonic) and/or the nutritive content of the food. Regarding the former, lateral ventricle or direct ACBsh MCH administration enhances the positive hedonic orosensory response[29]. For the latter, future work is needed to determine if MCH signaling enhances different nutritive and absorptive aspects of eating to drive consummatory behaviors early in the course of a meal. More specifically, it would be interesting to evaluate the extent that nutrient modulation of MCH neuron activity is correlated with levels of peripheral metabolites.

While recording of physiological MCH neuron Ca²⁺ activity and selectively activated MCH neurons during appetitive and consummatory behaviors provide complementary approaches that together inform about the endogenous role of these neurons, we note that there are two evident limitations of this study. First, this study does not evaluate the loss of MCH neuron function during these behaviors. Second, only male rats were used in this study, and given that the MCH system has previously been shown to have sex-specific effects on feeding[17,33], such future evaluation of potential sex differences in MCH neuron mediation of food cues is warranted.

Central MCH signaling promotes caloric consumption and overall positive energy balance in rodents[34–37] and, therefore, has been increasingly targeted as a potential pharmaceutical target for obesity treatment. Our collective results extend knowledge of the MCH system by suggesting that MCH neurons increase food consumption by enhancing the reinforcing effects of both preprandial appetitive and early prandial appetition processes. Future work is needed to decipher the prandial ingestive stage(s) (e.g., orosensory flavor, post-oral gustatory, post-oral nutritive) through which MCH neurons enhance food consumption.

## Methods

### Animals

Based on studies that have shown that the endogenous estrous stage is a critical determinant of MCH effects on eating in females[17,33], we have chosen to use male rats for this manuscript. For all experiments, male Sprague-Dawley rats (Envigo, Indianapolis, IN) weighing 300–400 g were used and individually housed in shoebox cages. Except where noted, rats were given ad libitum access to chow (Rodent Diet 5001, LabDiet, St. Louis, MO) and water. Rats were housed in a 12 h:12 h reverse light/dark cycle (lights off at 10:00 a.m.) or a 12 h:12 h light/dark cycle (lights off at 6:00 p.m.). All experiments were performed in accordance with NIH Guidelines for the Care and Use of Laboratory Animals, and all procedures were approved by the Institutional Animal Care and Use Committee of the University of Southern California.

### Stereotaxic optic fiber implantation

Surgeries were adapted from procedures described previously[38]. Rats were first anesthetized and sedated via intramuscular injection of a

ketamine (90 mg/kg), xylazine (2.8 mg/kg), and acepromazine (0.72 mg/kg) cocktail, prepped for surgery, and placed in stereotaxic apparatus. They were given subcutaneous injections of buprenorphine SR (0.65 mg/kg) during surgery as an analgesic. A fiber-optic cannula (Doric Lenses Inc, Quebec, Canada; flat 400-μm core, 0.48 numerical aperture (NA) was implanted in the LHA at the following coordinates:[39] −2.9 mm anterior/posterior (AP), +1.6 mm medial/lateral (ML), −8.6 mm dorsal/ventral (DV) (0 reference point at bregma for ML, AP, and DV). The optic fiber was then affixed to the skull with jeweler's screws, instant adhesive glue, and dental cement. All subjects were given 1 week to recover from surgery prior to experiments.

## Intragastric (IG) catheter implantation

Gastric catheter surgeries were performed and adapted from procedures described previously[40,41]. Following an overnight fast, rats were anesthetized with a ketamine (90 mg/kg), xylazine (2.8 mg/kg), and acepromazine (0.72 mg/kg) cocktail and then laparotomized while under isoflurane (~5% induction rate; ~1.5–3% maintenance rate). A gastric catheter made of silastic tubing (inside diameter = 0.64 mm, outside diameter = 1.19 mm; Dow Corning, Midland, MI) was inserted through a puncture wound in the greater curvature of the forestomach. Importantly, the tip of the tubing was fitted with a small silastic collar (inside diameter = 0.76 mm, outside diameter = 1.65 mm; Dow Corning, Midland, MI) that served as an anchor to keep the tube in the stomach and secured via a purse-string suture. The catheter was then fixed against the stomach with a single stay suture and a small piece of square Marlex mesh (Davol, Cranston, RI). A purse-string suture and concentric serosal tunnel were used to close the wound in the stomach. The other end of the catheter was then passed through an incision through the abdominal muscle and was tunneled subcutaneously to an interscapular exit site, where it was attached using a single stay suture and a larger square piece of Marlex mesh. The tube was then connected to a Luer lock adapter, as part of a backpack harness worn by the rat around-the-clock (Quick Connect Harness, Strategic Applications, Lake Villa, IL). Rats were treated postoperatively with gentamicin (8 mg/kg sc) and ketoprofen (1 mg/kg sc). Rats were given increasing increments of chow (1–3 pellets) after surgery and then ad libitum access to chow. The gastric catheter was routinely flushed with 0.5 ml of isotonic saline beginning 48 h after surgery to maintain its patency. Harness bands were adjusted daily to accommodate changes in body mass.

## Immunohistochemistry

Rats were first anesthetized and sedated via intramuscular injections of ketamine (90 mg/kg), xylazine (2.8 mg/kg), and acepromazine (0.72 mg/kg) cocktail and then perfused using 0.9% sterile saline (pH 7.4), followed by 4% paraformaldehyde (PFA) in 0.1 M borate buffer (pH 9.5; PFA). Brains were extracted and post-fixed with 12% sucrose in PFA overnight, and then flash-frozen in methyl-butane cooled by dry ice. Brains were sectioned into 30-μm sections on a microtome cooled with dry ice and collected in an antifreeze solution, and stored in a −20 °C freezer.

The following IHC fluorescence labeling procedures were adapted from previous work[22,42]. Rabbit anti-MCH (1:5000; PhoenixPharmaceuticals, Burlingame, CA, USA; Catalog #: H-070-47; Clonality: Polyclonal; Lot #: 46317) and rabbit anti-RFP (1:2000, Rockland Inc., Limerick, PA, USA; Catalog #:600-401-379; Clonality: Polyclonal) were the two antibodies used. Antibodies were prepared in 0.02 M potassium phosphate-buffered saline (KPBS) solution containing 0.2% sodium azide and 2.0% normal donkey serum and stored at 4 °C overnight. After many series of washing with 0.02 M KPBS, brain sections were incubated in a secondary antibody solution. The two secondary antibodies used, donkey anti-rabbit AF647 (Catalog #: 711-606-152; Lot #: 160172) and donkey anti-rabbit AF488 (Catalog #: 711-546-152; Lot #: 126798), had a 1:500 dilution and were stored overnight

at 4 °C (Jackson Immunoresearch; West Grove, PA, USA). Sections were then mounted and coverslipped using 50% glycerol in 0.02 M KPBS and clear nail polish was used to seal the coverslip onto the slide.

Antibody tagging of MCH first involved washing the brain sections on a motorized rotating platform in the following order (overnight incubations on a motorized rotating platform at 4 °C): (1) 0.02 M KPBS (change KPBS every 5 min for 30 min), (2) 0.3% Triton X-100 in KPBS (30 min), (3) KPBS (change KPBS every 5 min for 15 min), (4) 2% donkey serum in KPBS (10 min), (5) 2% normal donkey serum, 0.2% sodium azide, and rabbit anti-MCH antibodies [1:2000; rabbit anti-MCH] in KPBS (-24 h)[43] (6) KPBS (change KPBS every 10 min for 1 h), (7) 2% normal donkey serum, 0.2% sodium azide, and secondary antibodies (1:1000; donkey anti-rabbit AF647, Jackson Immunoresearch; overnight) in KPBS (-30 h), (8) KPBS (change KPBS every 2 min for 4 min). Sections were then mounted, air-dried, and coverslipped with 50% glycerol in a 0.02 M KPBS mounting medium. Photomicrographs were acquired using a Nikon 80i (Nikon DS-QI1,1280 × 1024 resolution, 1.45 megapixel) microscope under epifluorescence.

## Intracranial virus injection

Rats were first anesthetized and sedated via intramuscular injections of ketamine (90 mg/kg), xylazine (2.8 mg/kg), and acepromazine (0.72 mg/kg) cocktail, prepped for surgery, and placed in stereotaxic apparatus. Stereotaxic injections of viruses were delivered using a micro-infusion pump (Harvard Apparatus, Cambridge, MA, USA) connected to a 33-gauge microsyringe injector attached to a PE20 catheter and Hamilton syringe. The flow rate was calibrated and set to 5 μl/min. Injectors were left in place for 2 min post-injection. Following viral injections, animals were either implanted with an optic fiber or surgically closed with sutures or skin glue. Experiments occurred either 21 days after virus injection to allow for virus transduction and expression (MCH DREADDS) or when animals showed viable fluorescence signals (GCAMP6s with MCH promoter). Successful virally mediated transduction was confirmed postmortem in all animals via IHC staining using immunofluorescence-based antibody amplification to enhance the fluorescence transgene signal, followed by manual quantification under epifluorescence illumination using a Nikon 80i (Nikon DS-QI1,1280 × 1024 resolution, 1.45 megapixel). The MCH DREADDS virus is now commercially available from Vector Biolabs (Malvern, PA, USA) upon request, and MCH GCAMP6s is available from the authors upon reasonable request.

For the recording of MCH neuron $Ca^{2+}$ activity, 1 μl of an AAV9.pMCH.GCaMP6s.hGH (MCH promoter-driven GCaMP6s) was unilaterally injected at the following coordinates:[39] −2.9 mm AP, +1.6 mm ML, −8.8 mm DC (0 reference point for AP, ML, and DV at bregma). An optic fiber was implanted (−2.9 mm AP, +1.6 mm ML, −8.6 mm DV) above the injection site as described before.

For chemogenetic activation of MCH neurons via DREADDS, an AAV2-rMCHp-hM3D(Gq)-mCherry (MCH DREADDs) was bilaterally injected at the following coordinates:[39] injection (1) −2.6 mm AP, ±1.8 mm ML, −8.0 mm DV; (2) −2.6 mm AP, ±1.0 mm ML, −8.0 mm DV; (3) −2.9 mm AP, ±1.1 mm ML, −8.8 mm DV; (4) −2.9 mm AP, ±1.6 mm ML, −8.8 mm DV (0 reference point for AP, ML, and DV at bregma) to target MCH neurons in the LHA and ZI. The injection volume was 200 nl/site.

## Characterization of MCH GCAMPS and DREADDs expression

Immunofluorescence colocalization of MCH and fluorescence reporter in MCH GCAMPs virus was conducted in sections from Swanson Atlas levels 28–32[39], based on IHC staining for MCH (as described above). All animals showed selective immunofluorescence colocalization, such that the fluorescence reporter was exclusive to neurons with the MCH tag. All animals were included in experimental analyses.

For MCH DREADDs experiments, staining for RFP to amplify the mCherry signal was conducted as described above. Counts were

performed in sections from Swanson Brain Atlas level 28–32[39], which encompasses all MCH-containing neurons in the LHA and ZI. For MCH DREADD experiments, animals were excluded from all experimental analyses if fewer than 2/3 of the total number of MCH neurons were transduced with RFP (based on IHC staining for MCH). All animals met these criteria and were included for experimental analyses.

For MCH GCAMPs quantification, GCAMPs6 expression was quantified in one out of five series of brain tissue sections from the perfused brains cut at 30 μm on a freezing microtome based on counts for the fluorescence reporter GFP. Counts were performed in sections from Swanson Brain Atlas level 27–32[39], which encompasses all MCH-containing neurons. Sections were stained for MCH+ neurons and cell counts were performed in four GCAMP6s virus-injected animals. Researchers who performed the counting were kept consistent between cohorts and blind to experimental assignments.

### Drug preparation
For chemogenetic activation of MCH neurons, 1 ml/kg DCZ (100 μg/kg) or vehicle (1% DMSO in 99% saline) is administered intraperitoneally through a syringe. Doses and concentration of DCZ and vehicle were based on previous work[44]. Prior to drug administrations, animals were handled and prepared for injections.

Reagent-grade glucose (8 and 12%) was prepared fresh with $dH_2O$ as needed. Saccharin-sweetened Kool-Aid solutions were made by mixing either 0.05% unsweetened cherry or grape Kool-Aid powder with 0.01% sodium saccharin solution. Each solution was made fresh for each training or test day.

### In vivo fiber photometry
In vivo fiber photometry was performed according to previous work[45]. Photometry signal was acquired using the Neurophotometrics fiber photometry system (Neurophotometrics, San Diego, CA) at a sampling frequency of 40 Hz and administering alternating wavelengths of 470 nm ($Ca^{2+}$ dependent) or 415 nm ($Ca^{2+}$ independent). The fluorescence light is transmitted through an optical patch cord (Doric Lenses) and converges onto the implanted optic fiber, which in turn sends back neural fluorescence through the same optic fiber/patch cord and is focused onto a photoreceiver. All behaviors (cues, entries, eating bouts, etc.) were time-stamped using the data acquisition software (Bonsai). The resulting signals were then corrected by subtracting the $Ca^{2+}$ independent signal from the $Ca^{2+}$ dependent signal to calculate fluorescence fluctuations due to $Ca^{2+}$ (corrected signal) and not due to baseline neural activity or motion artifacts and fitted to a biexponential curve. The corrected fluorescence signal was then normalized within each rat by calculating the ΔF/F using the average fluorescence signal for the entire recording and converting the signal to z-scores. The normalized signal was then aligned to behavioral events of interest (cues, entries, eating bouts, etc.), and data extraction was done using the original MatLab code.

### Pavlovian Discrimination Task
Testing and training were adapted from previous procedures[46]. For this experiment, animals were housed in a reverse light/dark cycle (lights off at 10:00 a.m.). Animals were provided with overnight access to sucrose solution (11% weight/volume) in the home cage and were required to consume 50 ml of the solution to move onward with training. Animals were then chronically restricted to 15 g of chow daily (Rodent Diet 5001, LabDiet, St. Louis, MO, USA) and given chow after each session. Animals were placed in identical operant chambers (Med Associates), which contained an accessible lickometer filled with sucrose solution when activated. Each session consisted of eight conditioned-stimulus-positive (CS+) and eight conditioned-stimulus-negative (CS−) audio cues, where immediately after the CS+, the sucrose solution became accessible for 20 s, and after the CS−, no sucrose reward was provided. The order of the cues and the time

between each cue was random, with the average time between cues being around 110 s. The cues consisted of either a clicking noise or tone-like frequency and were counterbalanced across animals, for which was the CS+ or CS−. Each session was 45 min long and animals had seven total training sessions.

For the effects of physiological MCH neuron $Ca^{2+}$ activity during the Pavlovian Discrimination Task, using in vivo fiber photometry, a patch cord was connected to the implanted optical fiber, and LEDs were delivered, alternating between 470 nm ($Ca^{2+}$ dependent) and 415 nm ($Ca^{2+}$ independent) during the second, fifth, and seventh CS training sessions to assess the speed of cue response development.

For the effects of chemogenetic activation of MCH neurons on performance in the Pavlovian Discrimination Task, animals were randomized to receive either IP DCZ or vehicle using a counterbalanced (based on performance during training), within-subjects design with 72 h between treatments. IP injections of DCZ or vehicle occurred 5 min before the behavioral test.

### Conditioned place preference (CPP)
CPP training and testing procedures were conducted as described previously[19–21]. For this experiment, animals were housed in a reverse light/dark cycle (lights off at 10:00 a.m.). Briefly, the CPP apparatus consisted of two conjoined plexiglass compartments with a guillotine door separating the two sides (Med Associates, Fairfax, VT, USA). The two sides (contexts) were distinguished by wall color and floor texture. Rats were given a 15-min habituation session with the guillotine door open and video recording software (Anymaze) to measure time spent in each context. For each rat, the least preferred context during habituation was designated as the food-paired context for subsequent training. Training occurred in the early phase of the dark cycle and home cage chow was pulled 1 h prior to each training session. CPP training consisted of 12 (20 min, 5 days/week) sessions: six sessions isolated in the food-paired context and six sessions isolated in the non-food-paired context. Context training order was randomized. During the food-paired sessions, 5 g of 45% kcal high fat/sucrose diet (D12451, Research Diets, New Brunswick, NJ, USA) was placed on the chamber floor, and no food was presented during non-food-paired sessions. All rats consumed the entire 5 g of food during each food-paired session.

CPP testing occurred 2 days after the last training session and 1 h prior to the test session, home cage chow was removed. During testing, the guillotine door remained open and rats were allowed to freely explore both contexts for 15 min. No food was present during testing. Time spent in each context during the test was calculated from video recording software (Anymaze). For the effects of physiological MCH neuron $Ca^{2+}$ activity during the CPP, using in vivo fiber photometry, a patch cord was connected to the implanted optical fiber, and LEDs were delivered, alternating between 470 nm ($Ca^{2+}$ dependent) and 415 nm ($Ca^{2+}$ independent) during the task. In addition, a customized door (Viterbi/Dornsife USC machine shop) separating both contexts was made to ensure the optic fiber/patch cord connection could safely maneuver between contexts without interfering with the integrity of the experiment. For the effects of chemogenetic activation of MCH neurons on performance in CPP, animals were randomized to receive either IP DCZ or vehicle using a counterbalanced (based on performance during training), within-subjects design with 72 h between treatments. IP injections of DCZ or vehicle occurred 5 min before the behavioral test.

### Refeeding after an overnight fast
Animals were housed in reverse light/dark cycle (lights off at 10:00 a.m.). Prior to test day, animals were overnight fasted but had access to water. On test days, animals were exposed to a neutral context, where they had 15 min access to the context with no food, followed by a 30-min chow (Laboratory Rodent Diet 5001, St. Louis, MO, USA) access period and then a 10-min

post-food access period, where the food is removed. For the effects of physiological MCH neuron $Ca^{2+}$ activity during the refeeding after an overnight fast, using in vivo fiber photometry, a patch cord was connected to the implanted optical fiber, and LEDs were delivered, alternating between 470 nm ($Ca^{2+}$ dependent) and 415 nm ($Ca^{2+}$ independent) during the task. For the effects of chemogenetic activation of MCH neurons on performance on refeeding after an overnight fast, animals were randomized to receive either IP DCZ or vehicle using a counterbalanced (based on performance during training), within-subjects design with 72 h between treatments. IP injections of DCZ or vehicle occurred 5 min before the behavioral test.

### Pavlovian-instrumental transfer (PIT)

The paradigm was adapted from previous procedures[23,24]. For this experiment, animals were housed in a reverse light/dark cycle (lights off at 10:00 a.m.). This behavioral procedure consists of four phases: Pavlovian discrimination conditioning, instrumental conditioning, instrumental extinction, and PIT test day. Each stage occurred in identical operant chambers (Med Associates), where each chamber was equipped with a retractable lickometer, retractable levers on the opposite sides of the chamber, and speakers to emit audio cues. The procedure for Pavlovian training is the same as the Pavlovian Discrimination Task described above.

For instrumental conditioning, animals were placed in identical operant chambers (Med Associates) containing a retractable lickometer filled with sucrose solution and a retractable lever on the opposite side of the chamber. When the lever is pressed, the lickometer becomes accessible for 20 s. Animals received five sessions where the lever remained retracted unless pressed. After eight presses of the lever or 20 min, the session was over. For the next 12 sessions, the lever became accessible at random intertrial intervals (average 110 s between trials), consisting of eight opportunities to press the lever. If the animals did not press the lever within 30 s of it being retracted, the lever detracted and the next trial began. Each of these sessions was 45 min.

For instrumental extinction, animals were placed in identical operant chambers (Med Associates) and received similar instrumental conditioning, except that when the lever became accessible at random intertrial intervals (average 110 s), pressing the lever did not result in access to a lickometer filled with sucrose solution. Each session consisted of eight opportunities to press the lever, lasting 45 min, and they received 12 such sessions.

For PIT test day, the effect of the Pavlovian stimuli on instrumental behavior (lever pressing) was evaluated during the transfer test, where animals were placed in identical operant chambers (Med Associates) for 45 min. The test session consisted of eight CS+ and eight CS− audio cues, the same ones used during the Pavlovian discrimination training, followed immediately by the lever being accessible for each audio cue. Pressing the lever did not result in a sucrose reward, regardless of which audio cue was played prior. For the effects of chemogenetic activation of MCH neurons on performance during PIT, animals were randomized to receive either IP DCZ or vehicle using a counterbalanced (based on performance during training), within-subjects design with 72 h between treatments. IP injections of DCZ or vehicle occurred 5 min before the behavioral test.

### Flavor preference conditioning

Animals were housed in a light/dark cycle (lights off at 6:00 p.m.). After rats successfully recovered from surgery (IG catheter implantation and intracranial injection of excitatory MCH DREADDS), they were food-restricted to maintain 85% of their current body weight (fed rations daily after lights out). To acclimate the rats to the lickometers, they were overnight water-deprived and given a 1-h session to drink 8% glucose from the lickometer, while receiving IG 8% glucose infusions.

To acclimate the rats to the saccharin-sweetened solution, rats were given two 30-min sessions on separate days with unflavored 0.1% saccharin-sweetened solution with no IG infusions.

Following this acclimation period, rats were trained to lick two different saccharin-sweetened Kool-Aid flavors, being exposed to each flavor for six sessions. Animals are only exposed to one flavor per training session and each flavor is paired with IG glucose. Prior to each training session, using a counterbalanced, within-subjects design, animals were injected with IP DCZ (to activate MCH neurons) or vehicle, dependent on the flavor they were exposed to that day. The flavor paired with DCZ is the conditioned-stimulus positive (CS+), while the vehicle-paired flavor is the conditioned-stimulus negative (CS−). Animals were water-deprived overnight and given 1-h access to lickometer each session, and IG glucose was infused each time they licked. In addition, for each training session, animals were capped at 1499 licks, to ensure that the same amount of glucose was infused for the CS− and CS+. Animals were given two test days (pre- and post-training) where they had access to both flavors at once. During test days, there were no treatments, infusion of nutrients, or overnight water deprivation. The lesser preferred flavor during the first test day (pre-test) was paired with DCZ during the training sessions. The number of licks per flavor was recorded for each training and test session and used as a measure to indicate preference.

### Food intake studies

Animals were housed in a reverse light/dark cycle (lights off at 10:00 a.m.). Home cage chow (Rodent Diet 5001, LabDiet, St. Louis, MO, USA) was removed 2 h prior to the light onset (10am). For chemogenetic activation of MCH neurons, animals were counterbalanced, using a within-subjects design to receive IP DCZ or vehicle 5 min prior to light onset, and pre-weighed amounts of the test chow diet were deposited in the home cage immediately after the light onset. The same procedure for evaluating the effects of IP DCZ without MCH DREADDs. Spill papers were placed underneath the cages to collect food crumbs. Food spillage was weighed and added to the difference between the initial hopper weight and the hopper weight at each measurement time point. A total of 72 h was allotted between treatments

### Statistical analyses

Statistical analyses were performed using GraphPad Prism 9.0 software (GraphPad Software Inc., San Diego, CA, USA) and Microsoft Excel V16.66.1. Data are expressed as mean ± SEM. Statistical details can be found in the figure legends and $n$'s refer to the number of animals for each condition. Differences were considered statistically significant at $P < 0.05$.

A two-tailed paired Student's $t$-test was used to compare MCH neuron $Ca^{2+}$ activity for the CS− and CS+ during the Pavlovian Discrimination Task, for the unpaired and paired contexts during CPP and for within eating bout and interbout intervals and 5 min pre-food and post-meal periods during refeeding. Simple linear regression analysis assessed a correlation between MCH $Ca^{2+}$ activity and latency to lick for sucrose solution in the Pavlovian Discrimination Task, entrance into the paired side of the CPP, and chow intake, meal period and bout duration during refeeding.

Two-tailed paired Student's $t$-tests were also used to compare vehicle to DCZ conditions in the Pavlovian Discrimination Task, PIT, and CPP, as well as to compare CS− versus CS+ in the flavor preference conditioning. A two-tailed unpaired Student's $t$-test was used to compare vehicle to DCZ conditions during refeeding. Two-way ANOVA with repeated measures and multiple comparisons were used to compare home cage chow intake under vehicle vs. DCZ conditions. Outliers were identified using the Grubb's test for

outliers post-hoc at signficane level of alpha = 0.05. For all experiments, assumptions of normality, homogeneity of variance (HOV), and independence were met where required.

One-way ANOVAs with repeated measures were used to assess differences in training data for the Pavlovian Discrimination Task and PIT.

## Reporting summary

Further information on research design is available in the Nature Portfolio Reporting Summary linked to this article.

## Data availability

All data generated and analyzed for this manuscript are available from the corresponding senior author (S.E.K.) upon reasonable request. The source data underlying Figs. 1d–f, h–k, 2c–h, 3d–j, 4d–o, i–l, 5c–e, g and 6b–d; Supplemental Fig. 1b–f, 2a–c, 3a, b, 4b, d, 5a–c and 6a–c, e, g are provided as a source data file with this paper. The data from this manuscript are available in the Open Science Framework Repository https://doi.org/10.17605/OSF.IO/WMKGJ upon the date of publication. Source data are provided with this paper.

## Materials availability

Unique biological materials (AAVs containing excitatory DREADDS with MCH promoter) are commercially available from Vector Biolabs. The MCH GCAMP6s virus is available from the corresponding senior author (S.E.K.) upon reasonable request.

## Code availability

All codes generated for this manuscript are available from the corresponding senior author (S.E.K.) upon reasonable request. The original code and a demo file are available in the Open Science Framework Repository https://doi.org/10.17605/OSF.IO/WMKGJ upon the date of publication.

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

## Acknowledgements
This work was supported by a National Science Foundation Graduate Research Fellowship to K.S.S., National Institute of Diabetes and Digestive and Kidney Diseases grants: DK104897 and DK123423 to S.E.K. and DK118000 and DK128306 to E.E.N., a Postdoctoral Ruth L. Kirschtein National Research Service Award from the National Institute of Aging (F32AG077932) to A.M.R.H. and a Quebec Research Funds postdoctoral fellowship (315201) and an Alzheimer's Association Research Fellowship to Promote Diversity (AARFD-22-972811) to L.D.S., an NIH RO1 (DC018562) to L.A.S., and ETH Zürich funded D.B. Clozapine-N-Oxide was kindly provided by the National Institute of Mental Health. The authors are grateful to the Kanoski lab undergraduate research assistants for their assistance in behavioral experiments and histology.

## Author contributions
K.S.S. and S.E.K. conceived and performed experiments and wrote the manuscript. L.T.L., A.M.R.H., L.D.S., K.M., K.N.D., A.E.K., and A.G.B. performed experiments. A.C.N. helped with revision experiments. D.B. provided expertise and virus. E.E.N. and L.A.S. provided expertise.

## Competing interests
The authors declare no competing interests.
