## [Peer Review File · Nature Communications]

Hypothalamic melanin-concentrating hormone neurons integrate food-motivated appetitive and consummatory processesREVIEWER COMMENTS

Reviewer #1 (Remarks to the Author):

The neural basis for the brain to integrate appetitive and consummatory signals is a long-standing question that remains to be addressed. In this manuscript entitled "Hypothalamic melanin-concentrating hormone neurons integrate food-motivated appetitive and consummatory processes," Subramanian et al. provide evidence that the lateral hypothalamic area (LHA) melanin-concentrating hormone (MCH) releasing neurons activity is responsible for the integration of both food-directed appetitive and consummatory processes in rat. First, using Ca²⁺-signal photometry in the LHA of rats injected with AAV-mediated MCH promoter-driven expression of GCaMP6s in sucrose-predictive Pavlovian tasks, they reveal MCH neuron Ca²⁺ activity increases in response to both discrete and contextual food predictive cues and is correlated with food-motivated responses. Second, they used the conditioned Place Preference (CPP) paradigm to reveal that MCH neurons activity increases in response to contextual-based food-predictive cues and is associated with food-seeking memory behavior. Third, in free-moving animals, they found MCH neuron activity increases during active eating and is highly predictive of cumulative caloric intake amounts and eating bout durations within a meal. Fourth, they found that chemogenetic activation of MCH neurons increases appetitive responses to discrete food-predictive cues and positively correlates with meal sizes. Finally, activation of MCH neurons promotes preferences for non-caloric flavor when paired with intragastric glucose infusion. Overall, the study addressed a long-standing question in the field; the data presented are convincing, and the conclusions are justified. The paper is clearly written. I have a few comments that may help improve the impact of the study.

1. While it appears statistically insignificant, there was a prominent peak of Ca²⁺ signal elevations pre-initiation of the CS+ sound (Fig. 1g, i). Are there any explanations? A heatmap of Ca²⁺ level changes as a function of time for each trial/animal is a better way to show the data (e.g., see data presentation in PMID:31277924). Moreover, please clarify what "n=7" means in the figure legend. 7 animals or 7 trials? It seems there is a second peak at 25s. Was it because the animal started licking? However, the latency-to-lick appears to be less than 10 s (Fig. 1e). Please comment on this.

2. It is intriguing that in the CPP paradigm, the author showed chemogenetic activation of MCH neurons increased time spent on food paired side. As we know, chemogenetic activation of neurons in vivo lacks time resolution (initiation, activation levels, and durations). Optogenetic activation using the restraint paradigm from figure 2 and figure 5 would be better to resolve this issue. Or maybe the authors can show a time course of Ca²⁺ level changes in these neurons after injecting DREADD agonists. c-fos immunohistochemistry (maybe co-stain with MCH) or electrophysiology recording will also help verify the chemogenetic DREADD activation.

3. In figure 3, a heatmap is required for fiber photometry data presentation (similar to that shown in PMID: 31277924). Please clarify the n number shown in the figure. Moreover, I wondered when the MCH neuronal firing (Ca²⁺-elevation) would return to normal after the meal. Is it possible for the authors to show one representative trace using $\Delta F/F$ for the photometry data?

4. The data presented in this study shows strong evidence that the activation of MCH neurons enhances

the valuation of nutrient and appetitive processes. An additional inhibitory DREADD experiment may improve the illustration of whether MCH neurons are the critical integrators for appetite and ingestive behavior.

5. Whether MCH-signaling is involved in food-motivated appetitive and intake-promoting consummatory processes is not yet clear. Maybe the authors can elaborate a little bit on the underlying mechanisms in the discussion section.

Reviewer #2 (Remarks to the Author):

This study by Subramanian et al., explored relation of MCH neuron activity with appetition. They used in vivo activity imaging and chemogenetic gain of function to show that MCH neurons respond to food related cues and the magnitude of this response correlates with consumption. Furthermore, gain of function studies extended earlier work on the sufficiency of MCH neuron activity in promoting appetitive behavior. Overall, the experiments are well designed and nicely executed. The results are solid and supports the conclusion. While the results are noteworthy, they can be strengthened with the following few suggestions.

Major points

1. The main weakness of the study is that the manipulations mainly rely on gain of function but does not address whether MCH neuron activity is necessary for appetitive response under physiological conditions.
2. Imaging studies were beautifully performed but there is a missed opportunity here as to whether and how metabolic hormones, such as leptin, affect these MCH neuronal responses to food predicting cues.
3. Another missed opportunity is to monitor development of MCH response to food predictive cues over the training trials. How many training sessions does it take? Does the speed of cue response development depend on caloric value of the reward?

Minor points

1. Example trace in fig 3c does not represent what is described in the text. Interbout activity appears to stay as high as during bout activity. Do the authors quantify peak z-score or average activity?
2. Line 105 - Ca²⁺ should be superscript.
3. Line 112 Fig I should be Fig 1i
4. Line 270 Prolongue should be prolong

Reviewer #3 (Remarks to the Author):

This study by Subramanian and colleagues examines the role of melanin-concentrating hormone (MCH) in food-motivated appetitive and consummatory processes. The authors find, using in vivo fiber photometry in male rats, that MCH neuron Ca²⁺ activity increases in response to both discrete and

contextual food predictive cues and also increases during eating and declines throughout a meal. They also find, using MCH-specific Gq DREADDs in male rats, that MCH neuron activation promotes appetitive behavioral responses to food-predictive cues, increases meal size, and enhances preference for a noncaloric flavor paired with intragastric glucose. The authors conclude that MCH neurons integrate appetite and appetition, an early meal positive feedback process that promotes further consumption.

This is a novel set of experiments that use cutting-edge techniques to measure and manipulate MCH neuron activity across an array of behavioral paradigms. The conclusions are largely supported by the data and should be of great interest to the feeding field. My major concern is that only males were used for these studies; however, I have a number of additional specific issues with the current submission.

Specific Issues:

1. The term “appetition” should be integrated into the title, since this is the major, specific focus of this manuscript.
2. General comment: Since viral vectors were used to allow for activation of MCH neurons, the correct term is "transduction", not "transfection". This should be corrected throughout the manuscript.
3. Experiment 1 (lines 101 – 102): The authors report that GCaMP6 signal was exclusive to MCH+ neurons, but what was the percentage of MCH+ neurons that co-labeled with GCaMP6 and did this vary by region within the hypothalamus (lateral hypothalamus, perifornical area, zona incerta)?
4. Experiment 1 (lines 106 – 108): No statistical tests appear to have been applied to support the statement that “The animals readily learned the Pavlovian discrimination, exhibited as increased number of licks per trial, reduced latency to lick, and increased number of CS+ trials with a consummatory response”.
5. Experiment 2: While I understand how a case could be made that conditioned place preference engages contextual cues associated with food, this paradigm is more commonly thought of as a test of liking of a stimulus (in this case, food). The authors should more clearly explain how conditioned place preference engages food-predictive Pavlovian context cues.
6. Experiment 3, Figure 3 c: There appears to be a spike in MCH neuron Ca²⁺ activity shortly after voluntary meal termination – why? It would be helpful to track glucose levels in parallel with eating activity, to determine if glucose might be directly responsible for the fluctuations in MCH neuron Ca²⁺ activity.
7. Experiment 4 (lines 176 – 179): This sentence requires editing for clarification. Also, DREADDs should not be in quotation marks.

8. Experiment 4 (lines 189 – 191, 198 – 203): No statistical tests appear to have been applied to support the statements that “...animals successfully learned the Pavlovian discrimination” and “... Animals showed increased licking (etc)” and “...animals increased their latency to lever press (etc)”.

9. Experiment 5: Did chemogenetic activation of MCH neurons during CPP testing also affect locomotor activity? This could be a potential confound for the finding that activation of MCH neurons increased the percentage of time spent on the food-paired context. The authors should provide measures of locomotor activity during CPP testing.

10. Experiment 7 (lines 250 – 252, Figures 6 b-c and S6 a-b): The baseline number of licks appears to be very different between MCH and control DREADDs groups. Why?

11. Methods, Animals: Only in this section is it addressed that only male rats were used for these studies. The NIH policy on Sex as a Biological Variable is that “NIH expects that sex as a biological variable will be factored into research designs, analyses, and reporting in vertebrate animal and human studies.” Why were females not included in the present studies?

12. Methods: Animals were housed in a reverse light/dark cycle or a (regular) light/dark cycle, but in most cases, it is not clear in the Methods which housing condition and light cycle stage was used for which experiments and why. In light of the influence of the circadian cycle on MCH neuron activity, the housing condition and stage of the light-dark cycle at testing should be made explicit in the Methods for each experiment.

Response to Reviewers' Comments

Reviewer #1 (Remarks to the Author):

The neural basis for the brain to integrate appetitive and consummatory signals is a long-standing question that remains to be addressed. In this manuscript entitled "Hypothalamic melanin-concentrating hormone neurons integrate food-motivated appetitive and consummatory processes," Subramanian et al. provide evidence that the lateral hypothalamic area (LHA) melanin-concentrating hormone (MCH) releasing neurons activity is responsible for the integration of both food-directed appetitive and consummatory processes in rat. First, using Ca^{2+} -signal photometry in the LHA of rats injected with AAV-mediated MCH promoter-driven expression of GCaMP6s in sucrose-predictive Pavlovian tasks, they reveal MCH neuron Ca^{2+} activity increases in response to both discrete and contextual food predictive cues and is correlated with food-motivated responses. Second, they used the conditioned Place Preference (CPP) paradigm to reveal that MCH neurons activity increases in response to contextual-based food-predictive cues and is associated with food-seeking memory behavior. Third, in free-moving animals, they found MCH neuron activity increases during active eating and is highly predictive of cumulative caloric intake amounts and eating bout durations within a meal. Fourth, they found that chemogenetic activation of MCH neurons increases appetitive responses to discrete food-predictive cues and positively correlates with meal sizes. Finally, activation of MCH neurons promotes preferences for non-caloric flavor when paired with intragastric glucose infusion. Overall, the study addressed a long-standing question in the field; the data presented are convincing, and the conclusions are justified. The paper is clearly written. I have a few comments that may help improve the impact of the study.

We thank the reviewer for the careful and thorough review of our manuscript and for highlighting the importance and the novelty of our findings. Our responses are detailed below:

1. While it appears statistically insignificant, there was a prominent peak of Ca^{2+} signal elevations pre-initiation of the CS+ sound (Fig. 1g, i). Are there any explanations? A heatmap of Ca^{2+} level changes as a function of time for each trial/animal is a better way to show the data (e.g., see data presentation in PMID:31277924). Moreover, please clarify what "n=7" means in the figure legend. 7 animals or 7 trials? It seems there is a second peak at 25s. Was it because the animal started licking? However, the latency-to-lick appears to be less than 10 s (Fig. 1e). Please comment on this.

We also had noticed the peak in MCH neuron Ca^{2+} activity prior to CS+ presentation. We do not have a clear explanation for this elevation, but we note that this elevation is not statistically significantly different than the CS-. However, in response to Reviewer #2 (comment #3, below), for this revised submission we developed a new cohort of animals to record MCH neuron Ca^{2+} activity during the same Pavlovian Discrimination Task, but at different timepoints throughout training. From this systematic replication, the same statistically significant spike in activity during the CS+ remains, but

there is no longer a prominent peak pre-presentation of the CS+. Therefore this suggests that the prominent peak of activity pre-initiation of the CS+ seen in our previous submission was likely an anomaly and not biological. Updated data from this systematic replication, without a nonsignificant pre-CS+ peak activity anomaly, is now depicted in Figure 1.

We appreciate your suggestion regarding the heatmap. We now include in our updated Figure 1 a heatmap of the MCH neuron Ca^{+2} signal for each animal during each of the five seconds of the CS presentation (Fig. 1h).

In the figure legend, n=7 is referring to the number of animals in the cohort. There were 16 total CS trials (8 CS+, 8 CS-) per training/test session. The legend for Figure 1 has been updated to make this more clear. Furthermore, we speculate that the rise in MCH neuron Ca^{+2} activity around the 25s point is due to reduced licking by the end of the period and possibly due to the lickometer retracting (acting as a cue).

2. It is intriguing that in the CPP paradigm, the author showed chemogenetic activation of MCH neurons increased time spent on food paired side. As we know, chemogenetic activation of neurons in vivo lacks time resolution (initiation, activation levels, and durations). Optogenetic activation using the restraint paradigm from figure 2 and figure 5 would be better to resolve this issue. Or maybe the authors can show a time course of Ca^{2+} level changes in these neurons after injecting DREADD agonists. c-fos immunohistochemistry (maybe co-stain with MCH) or electrophysiology recording will also help verify the chemogenetic DREADD activation.

The DREADD ligand, Deschloroclozapine (DCZ; PMID: 32632286) at 100 μ g/kg, reaches max concentration in the brain after 5 minutes post I.P. injection in rodents, and remains at effective concentrations for 60 minutes, which spans the entirety of our CPP test procedure. Therefore, DCZ remains bioactive in the brain as a DREADD ligand with sufficient time resolution for our CPP procedures.

Furthermore, we are confident that excitatory DREADDs are functional in MCH neurons, not only based on present behavioral results, but we have also previously confirmed the temporal dynamics of the MCH-promoter driven excitatory DREADD using electrophysiology recordings of transfected coronal brain sections bathed in a similar DREADD ligand, CNO (PMID: 29861386). In addition, we have replicated our previously published functional effects of CNO in the present manuscript with I.P. DCZ by showing increase in home cage food intake following chemogenetic activation of MCH neurons with DCZ, and of comparable magnitude and timeframe as CNO (Supplemental Figure 4). Therefore, we believe that DCZ is able to sufficiently act as a DREADD ligand for MCH-promoter driven excitatory DREADDs for the behaviors assessed in this manuscript, and that additional electrophysiological and/or c-Fos analyses would not provide substantial benefit to the interpretation of present results.

3. In figure 3, a heatmap is required for fiber photometry data presentation (similar to that shown in PMID: 31277924). Please clarify the n number shown in the figure.

Moreover, I wondered when the MCH neuronal firing (Ca²⁺-elevation) would return to normal after the meal. Is it possible for the authors to show one representative trace using DeltaF/F for the photometry data?

According to your suggestion, we now include a heatmap representing each animal's MCH neuron Ca²⁺ signal during active eating bouts and interbout activity (Fig. 3d). Secondly, the "n" number refers to the number of animals in the experiment and the legend for Figure 3 has been update for clarity.

We only collected data for 5 minutes of MCH neuron Ca²⁺ activity post voluntary meal termination.

The trace in Figure 3c shows average activity (df/f), but the y-axis label says the units are in z-score. We have corrected this mistake and appropriately quantified the z-score for the representative trace. Thanks for pointing this out.

4. The data presented in this study shows strong evidence that the activation of MCH neurons enhances the valuation of nutrient and appetitive processes. An additional inhibitory DREADD experiment may improve the illustration of whether MCH neurons are the critical integrators for appetite and ingestive behavior.

Despite that we have successfully constructed and validated excitatory DREADDs viral constructs in MCH neurons (validated anatomically, physiologically, and functionally; e.g., PMID 32795460; 31664021; 29861386), our attempts over the past several years to validate inhibitory DREADDs in MCH neurons have not yielded successful viral constructs for this approach. We speculate, from our experience and multiple efforts on this front, that inhibitory DREADDs are not functional in MCH neurons in rats, thus precluding this tool from our current arsenal. We note, however, that the present study includes physiological recording of MCH neuron calcium activity, which like inhibitory DREADDs, gives functional insight into the endogenous role of this neural population. By combining this approach with MCH neuron-specific excitatory DREADDs and sophisticated behavioral analyses, we believe that our collective data set provides a powerful and multi-level evaluation of the role of MCH neurons in appetite and food cue responsivity. We nevertheless agree with you that, when feasible, loss-of-function approaches are useful, and thus in the Discussion section we now emphasize the inability to inhibit MCH neurons as a limitation of the project (lines 329-331).

5. Whether MCH-signaling is involved in food-motivated appetitive and intake-promoting consummatory processes is not yet clear. Maybe the authors can elaborate a little bit on the underlying mechanisms in the discussion section.

We now elaborate in the Discussion section on potential underlying MCH-signaling mechanisms to promote appetitive and consummatory processes (discussion section, lines 327-335)

Reviewer #2 (Remarks to the Author):

This study by Subramanian et al., explored relation of MCH neuron activity with appetite. They used in vivo activity imaging and chemogenetic gain of function to show that MCH neurons respond to food related cues and the magnitude of this response correlates with consumption. Furthermore, gain of function studies extended earlier work on the sufficiency of MCH neuron activity in promoting appetitive behavior. Overall, the experiments are well designed and nicely executed. The results are solid and supports the conclusion. While the results are noteworthy, they can be strengthened with the following few suggestions.

We thank the Reviewer for the positive feedback. Below we address each of your comments.

Major points

1. The main weakness of the study is that the manipulations mainly rely on gain of function but does not address whether MCH neuron activity is necessary for appetitive response under physiological conditions.

As indicated above to Reviewer #1, despite that we have successfully constructed and validated excitatory DREADDs viral constructs in MCH neurons (validated anatomically, physiologically, and functionally; e.g., PMID 32795460; 31664021; 29861386), our attempts over the past several years to validate inhibitory DREADDs in MCH neurons have not yielded successful viral constructs for this approach. We speculate, from our experience and multiple efforts on this front, that inhibitory DREADDs are not functional in MCH neurons in rats, thus precluding this tool from our current arsenal. We note, however, that the present study includes physiological recording of MCH neuron calcium activity, which like inhibitory DREADDs, gives functional insight into the endogenous role of this neural population. By combining this approach with MCH neuron-specific excitatory DREADDs and sophisticated behavioral analyses, we believe that our collective data set provides a powerful and multi-level evaluation of the role of MCH neurons in appetite and food cue responsivity. We nevertheless agree with you that, when feasible, loss-of-function approaches are useful, and thus in the Discussion section we now emphasize the inability to inhibit MCH neurons as a limitation of the project (lines 329-331).

2. Imaging studies were beautifully performed but there is a missed opportunity here as to whether and how metabolic hormones, such as leptin, affect these MCH neuronal responses to food predicting cues.

We thank for the reviewer for this exciting experimental follow-up suggestion, and we agree that this is important new direction in light of the present results. However, we believe that such an evaluation should be done thoroughly and would therefore take several years to complete and is thus beyond the scope of this paper. We believe that our present results, including results from new experiments suggested by you in this

revision (see below), are timely and are likely to stimulate such follow-up research in the field exploring MCH neuron interactions with various metabolic hormones.

3. Another missed opportunity is to monitor development of MCH response to food predictive cues over the training trials. How many training sessions does it take? Does the speed of cue response development depend on caloric value of the reward?

We agree that this is a very important question. To address this in this revised submission, we developed a new cohort of animals to record MCH neuron Ca^{+2} activity during the same Pavlovian Discrimination Task, but at different points during training. More specifically, we recorded activity during the second, fifth, and seventh training sessions to assess the speed of cue-induced neuron response development. During the second day of training, there were no differences in MCH neuron Ca^{+2} activity during CS+ vs. CS- presentation, but by day five of training, there were statistically significant increases in activity during the CS+ compared to CS- (Supplemental Figure 6, similar to Day 7 in Figure 1). In addition, these calcium imaging effects align nicely with the behavioral data, such that there were statistically significant increases in licks per trial, decreases in latency to lick, and increases in number of CS+ trials with a consummatory response by day 5 (but not by day 2) of training (Figure 1d-f). We thank the reviewer for this suggestion, as we believe that these new data now even more strongly support the hypothesis that MCH neuron activity is dynamically modulated by food-reinforced learning.

Minor points

1. Example trace in fig 3c does not represent what is described in the text. Interbout activity appears to stay as high as during bout activity. Do the authors quantify peak z-score or average activity?

The representative trace depicts MCH neuron Ca^{+2} activity during a refeeding after an overnight fast. The quantification for within bout activity is the difference in activity between the start and end of an eating bout (pink box), and interbout activity is the difference in activity after the end of an eating bout and before the start of the next eating bout (white boxes). While the activity overall remains high, we see a difference in the change in activity between within bouts and interbout periods. We have updated the legend for Figure 3 to clarify how within bout and interbout activity were quantified.

In addition, we apologize for the oversight of the trace in Figure 3c. The trace actually depicts average activity (df/f), but the y-axis label says the units are in z-score. We have corrected this mistake and appropriately quantified the z-score for the representative trace.

2. Line 105 - Ca^{2+} should be superscript.

Changed to " Ca^{+2} "

3. Line 112 Fig I should be Fig 1i

Changed to “Fig. 1i”

4. Line 270 Prolongue should be prolong

Changed to “prolong”

Reviewer #3 (Remarks to the Author):

This study by Subramanian and colleagues examines the role of melanin-concentrating hormone (MCH) in food-motivated appetitive and consummatory processes. The authors find, using in vivo fiber photometry in male rats, that MCH neuron Ca^{+2} activity increases in response to both discrete and contextual food predictive cues and also increases during eating and declines throughout a meal. They also find, using MCH-specific Gq DREADDs in male rats, that MCH neuron activation promotes appetitive behavioral responses to food-predictive cues, increases meal size, and enhances preference for a noncaloric flavor paired with intragastric glucose. The authors conclude that MCH neurons integrate appetite and appetition, an early meal positive feedback process that promotes further consumption.

This is a novel set of experiments that use cutting-edge techniques to measure and manipulate MCH neuron activity across an array of behavioral paradigms. The conclusions are largely supported by the data and should be of great interest to the feeding field. My major concern is that only males were used for these studies; however, I have a number of additional specific issues with the current submission.

We thank the Reviewer the positive feedback and for the suggestions. Below we respond to each of your comments.

Specific Issues:

1. The term “appetition” should be integrated into the title, since this is the major, specific focus of this manuscript.

Throughout the manuscript we emphasize that physiological MCH neuron Ca^{+2} activity is consistent with appetition processes, but we do not claim that it is driving or triggering appetition. Therefore, we believe that including “appetition” in the title would be overly suggestive of our claims in the manuscript, and we prefer a more cautious approach as we believe understanding the neural correlates of appetition will require decades of more targeted research in this area. We also believe that the word “appetition” has only recently gained serious traction in the field, and that its use in the title may not connect with the broad general readership of *Nature Communications*.

2. General comment: Since viral vectors were used to allow for activation of MCH neurons, the correct term is "transduction", not "transfection". This should be corrected throughout the manuscript.

When referring to viral vectors, we now use "transduction" instead of "transfection".

3. Experiment 1 (lines 101 – 102): The authors report that GCaMP6 signal was exclusive to MCH+ neurons, but what was the percentage of MCH+ neurons that co-labeled with GCaMP6 and did this vary by region within the hypothalamus (lateral hypothalamus, perifornical area, zona incerta)?

We thank the reviewer for this important question. Accordingly, we have now conducted additional rigorous quantification analyses that are now included in our revised submission. We found (and now report) that approximately $70.9 \pm 3.5\%$ of LHA MCH neurons and $18.9 \pm 1.1\%$ of perifornical area MCH neurons were colocalized with the MCH GCaMP6. In addition, of all MCH neurons infected with the MCH GCaMP6s virus, approximately $92.9 \pm 1.1\%$ were within the LHA. There were no MCH+ neurons in the zona incerta (ZI) infected with the MCH GCaMP6s virus, and the MCH GCaMP6s virus was exclusive to MCH+ neurons. These findings (lines 106-109) and methods for quantifying GCaMP6s expression (lines 697-703) are now included in the manuscript.

4. Experiment 1 (lines 106 – 108): No statistical tests appear to have been applied to support the statement that "The animals readily learned the Pavlovian discrimination, exhibited as increased number of licks per trial, reduced latency to lick, and increased number of CS+ trials with a consummatory response".

We have now included a one-way repeated measures ANOVA to depict that there were differences in licks, latency to lick and number of CS+ trials with a consummatory response over the course of training. We have updated the manuscript (including figures) to indicate these statistical tests (Fig. 1d-f), and we thank the reviewer for pointing this out.

5. Experiment 2: While I understand how a case could be made that conditioned place preference engages contextual cues associated with food, this paradigm is more commonly thought of as a test of liking of a stimulus (in this case, food). The authors should more clearly explain how conditioned place preference engages food-predictive Pavlovian context cues.

The CPP paradigm that we conducted develops place preference by contextual cue discrimination under sated conditions. Through conditioning, the rats associate a specific context with the opportunity to consume palatable food. On the test day, when they are given access to both contexts with no food present, they choose between spending time in the two locations.

We do not believe that the two interpretations are mutually exclusive. In other words, the presence of CPP under baseline conditions indicates that the animals “like”/prefer a context with palatable food available vs. one without. However, the *expression* of CPP during our testing procedures requires the animals to exhibit discrimination between the two locations based on previous learning about contextual cues (as no food is present during testing). That the testing conditions involve contextual cue discrimination independent of food consumption is now further emphasized throughout the manuscript, where appropriate (lines 134-137).

6. Experiment 3, Figure 3 c: There appears to be a spike in MCH neuron Ca²⁺ activity shortly after voluntary meal termination – why? It would be helpful to track glucose levels in parallel with eating activity, to determine if glucose might be directly responsible for the fluctuations in MCH neuron Ca²⁺ activity.

We do see an increase in MCH neuron Ca²⁺ activity across animals post voluntary meal termination in comparison to pre-meal activity (Figure 3f). MCH neurons have been extensively shown to be involved in regulating energy balance and we speculate that the elevation in activity post voluntary meal termination may be due to changes in energy status as a result of consuming a meal and the onset of regulatory processes to maintain this positive energy valence (i.e., rest and digest). In light of your suggestion, we now add additional discussion to this idea in the Discussion section (Lines 352-354).

We currently do not have the capacity to record glucose in real time while recording MCH neuron Ca²⁺ activity. We thank the reviewer for an interesting experiment suggestion for future studies, and we are hopeful that our timely results will stimulate such follow-up analyses from within the field.

7. Experiment 4 (lines 176 – 179): This sentence requires editing for clarification. Also, DREADDs should not be in quotation marks.

We have reworked the sentence to more accurately explain the MCH promoter-driven DREADD transfection in the LHA and ZI (results section, lines 187-190).

8. Experiment 4 (lines 189 – 191, 198 – 203): No statistical tests appear to have been applied to support the statements that “...animals successfully learned the Pavlovian discrimination” and “... Animals showed increased licking (etc)” and “...animals increased their latency to lever press (etc)”.

We have now included a one-way repeated measures ANOVA for each training phase of PIT to depict that animals have successfully learned each task. We have updated the manuscript to include the statistical tests (Fig. 4d-f,i-k).

9. Experiment 5: Did chemogenetic activation of MCH neurons during CPP testing also affect locomotor activity? This could be a potential confound for the finding that

activation of MCH neurons increased the percentage of time spent on the food-paired context. The authors should provide measures of locomotor activity during CPP testing.

We thank the reviewer for identifying the potential confound and we now report that there were no difference in locomotor activity due to chemogenetic activation of MCH neurons during CPP. We have updated Figure 5, accordingly, to include locomotor activity during CPP (Fig. 5e).

10. Experiment 7 (lines 250 – 252, Figures 6 b-c and S6 a-b): The baseline number of licks appears to be very different between MCH and control DREADDs groups. Why?

The variability between the baseline licks in these experiments can be attributed, in part, to the studies being done in different cohorts and one of the cohorts also being a few months older than the other. Secondly, after more carefully examining the Pre-CS-data represented in Figure 6b, we found that there is an outlier and we have now removed this animal from the experiment. When comparing the baseline licks between the two cohorts using a Welch's t test for unpaired groups with unequal sample sizes and variance there were no statistical differences, suggesting that this apparent difference in baseline licks is not statistically supported.

11. Methods, Animals: Only in this section is it addressed that only male rats were used for these studies. The NIH policy on Sex as a Biological Variable is that "NIH expects that sex as a biological variable will be factored into research designs, analyses, and reporting in vertebrate animal and human studies." Why were females not included in the present studies?

We agree that sex as a biological variable is very important, and our lab has several recent papers evaluating sex differences in feeding behavior (e.g., PMID 31728883; PMID 32795460; PMID 36546482). Of relevance to the present paper, we have recently shown that MCH neurons have sex-specific effects on food intake (PMID: 32795460) and this is consistent with previous literature indicating that pharmacological MCH orexigenic effects are more robust in males (PMID: 18191424). Both our study and this previous study show that endogenous estrous stage is a critical determinant of MCH effects on eating in females. For this reason we chose to establish the novel phenomena in the present study first in males. Our recently submitted NIH R01 proposal aims to extend these analyses to females, but to do so now would extend the manuscript by several years and would preclude the rapid communication of what we believe are timely and novel results. To address your comment in this revision, we now emphasize that using only males is an important limitation of this study that requires follow-up work (discussion section, lines 330-332).

12. Methods: Animals were housed in a reverse light/dark cycle or a (regular) light/dark cycle, but in most cases, it is not clear in the Methods which housing condition and light cycle stage was used for which experiments and why. In light of the influence of the circadian cycle on MCH neuron activity, the housing condition and stage of the light-dark cycle at testing should be made explicit in the Methods for each experiment.

We apologize for this oversight. The methods section has been updated to include the light cycle the animals were housed in for each experiment. In general, animals were housed in reverse light/dark cycle, as rats are more active when the lights are off and consume more of their daily intake during this period. In addition, all animals were individually housed in shoebox cages (now specified included in Methods → Animals).

REVIEWERS' COMMENTS

Reviewer #1 (Remarks to the Author):

Thanks for addressing the concerns I raised on the previous version of the manuscript. Congratulations on such a beautiful study! I have some minor concerns that you may want to take a look at:

1. Although “Ca⁺²” is being used, most of the time, we use “Ca²⁺” I believe in publications.
2. Figure 1i, “Ca+2” the “2+” should be superscript.
3. “The trace in Figure 3c shows average activity (df/f), but the y-axis label says the units are in z-score. We have corrected this mistake and appropriately quantified the z score for the representative trace.” It appears that the y-axis label is still “z-score.”

Reviewer #2 (Remarks to the Author):

The authors have sufficiently addressed my previous comments. While the persistent lack of any loss-of-function experiments, due to technical limitations, weakens overall conclusions, the study has sufficient novelty, provides new insights and directions into the MCH biology. Additionally, these limitations are now acknowledged. Therefore I support the publication of this manuscript.

Reviewer #3 (Remarks to the Author):

The authors have done an excellent job of addressing my concerns. Congratulations on your outstanding paper.

Response to Reviewers' Comments

Reviewer #1 (Remarks to the Author):

Thanks for addressing the concerns I raised on the previous version of the manuscript. Congratulations on such a beautiful study! I have some minor concerns that you may want to take a look at:

1. Although "Ca⁺²" is being used, most of the time, we use "Ca²⁺" I believe in publications.
2. Figure 1i, "Ca+2" the "2+" should be superscript.
3. "The trace in Figure 3c shows average activity (df/f), but the y-axis label says the units are in z-score. We have corrected this mistake and appropriately quantified the z score for the representative trace." It appears that the y-axis label is still "z-score."

We appreciate the addition suggestions to the manuscript and have addressed them below:

- 1) We have updated the manuscript by changing "Ca⁺²" to "Ca²⁺" throughout.
- 2) We updated all figures (Figures 1,2,3 and Sup. Fig 1) that have "Ca⁺²" to now be "Ca²⁺".
- 3) In the original submission, the trace in Figure 3c showed average activity in df/f (the y-axis scale was representing df/f, but the y-axis label was stating the trace is in "z-score"). In our resubmission, we updated the trace to accurately show activity in z-score (the y-axis scale is now representing z-score).